# Learning Disentangled Multi-Agent World Model for Decentralized Control

**Di Xue** [1 2]  **Jing Jiang** [1 2]  **Shaowei Zhang** [1 2]  **Wenhao Guo** [1 2]  **Lei Yuan** [1 2 3]  **Zongzhang Zhang** [1 2]  **Yang Yu** [1 2 3]

## Abstract

World models enable learning policies via latent imagination, offering benefits such as history compression and sample efficiency. However, applying world models to multi-agent tasks presents a fundamental challenge: modeling multi-agent dynamics in latent space requires integrating information from different agents, often creating spurious correlations between their latent states. Existing methods either operate in observation space, forgoing the benefits of latent representations; or rely on inter-agent communication, compromising fully decentralized execution. We present Disentangled Multi-Agent World Model (DMAWM), which utilizes independent agent modules to derive factorized latent states from local observations, and a shared environment module to model the joint latent states that align with the factorized structure. This design explicitly disentangles individual agent states, ensuring that imaginary rollouts faithfully simulate the constraints of decentralized execution while capturing multi-agent interactions. Empirically, DMAWM outperforms existing model-based and model-free approaches in convergence speed and final performance, with quantitative and qualitative analysis confirming that DMAWM learns disentangled latent representations suitable for decentralized execution.

## 1. Introduction

Model-based reinforcement learning (MBRL) has emerged as a highly effective approach in the single-agent domain. It has enabled superhuman performance in complex tasks such as Atari and board games without prior knowledge of the rules (Schrittwieser et al., 2020), and has allowed agents to master challenging objectives like collecting diamonds in Minecraft from scratch (Hafner et al., 2025). The cornerstone of these successes lies in learning models of the environments, often referred to as world models (Ha & Schmidhuber, 2018). World models enable learning policies via latent imagination, offering key benefits such as history compression into compact latent state representations and improved sample efficiency. These latent representations are predictive of future outcomes, enabling agents to efficiently plan or learn policies through simulated experience within the learned world model.

However, applying world models to multi-agent reinforcement learning (MARL) presents a fundamental challenge: modeling multi-agent dynamics in latent space requires integrating information from different agents, often creating spurious correlations between their latent states. Such correlations are impossible to maintain during decentralized execution, hindering the learning of disentangled latent states crucial for effective decentralized control. Existing model-based MARL methods have attempted to address these challenges with varying degrees of success. Some operate in the original observation space (Zhang et al., 2021; Xu et al., 2022; Zhang et al., 2025), forgoing the benefits of latent representations such as history compression and sample efficiency. Others rely on inter-agent communication to maintain the correlation during execution (Egorov & Shpilman, 2022; Wu et al., 2023; Toledo, 2024), compromising fully decentralized execution and failing to learn the disentangled latent states crucial for decentralized control.

To this end, we present the Disentangled Multi-Agent World Model (DMAWM), a novel framework designed to learn decentralized policies within a latent space. DMAWM adheres to the centralized training and decentralized execution (CTDE) paradigm, utilizing an architecture that comprises independent agent modules and a shared environment module. During decentralized execution, the independent agent modules derive factorized latent states from local observations; during centralized training, the shared environment module models joint latent states while being trained to align with the factorized structure. This design explicitly disentangles individual agent states, ensuring that imagined rollouts respect the constraints of decentralized execution

[1]National Key Laboratory for Novel Software Technology, Nanjing University, Nanjing 210023, China [2]School of Artificial Intelligence, Nanjing University, Nanjing 210023, China [3]Polixir Technologies, Nanjing 211106, China. Correspondence to: Zongzhang Zhang <zzzhang@nju.edu.cn>.

while capturing multi-agent interactions. Building on these disentangled representations, policies and value networks are trained on imagined trajectories and can be deployed in a fully decentralized manner. Additionally, we leverage real trajectories to ground value estimates, narrowing the gap between imagined and actual returns.

We evaluate DMAWM on three challenging MARL benchmarks, including SMAC (Samvelyan et al., 2019), SMACv2 (Ellis et al., 2023), and vision-based Melting Pot (Egorov & Shpilman, 2022). DMAWM outperforms existing model-based and model-free approaches in convergence speed and final performance. We further provide quantitative and qualitative analysis confirming that DMAWM learns disentangled latent representations suitable for decentralized execution.

## 2. Related Work

**World models for control.** World models learn the underlying dynamics from raw observations for data-efficient control. Generative world models are optimized to reconstruct the observation to capture the dynamics: SimPLe (Kaiser et al., 2020) trains a video predictor as an environment model, and the Dreamer family (Hafner et al., 2020; 2021; 2025) leverages recurrent state-space models (RSSMs) (Hafner et al., 2019) to plan in latent space. Implicit world models avoid decoding observations and instead couple learned dynamics with decision procedures—MuZero (Schrittwieser et al., 2020) and EfficientZero (Ye et al., 2021) integrate MCTS, while TD-MPC and TD-MPC2 (Hansen et al., 2022; 2024) pair implicit models with MPC for continuous control. Recent advances focus on stronger backbone models, bringing transformers and diffusion into dynamics learning: IRIS (Micheli et al., 2023) and TWM (Robine et al., 2023) apply transformers to model the whole trajectory in an end-to-end manner, while UniSim (Yang et al., 2024) and DIAMOND (Alonso et al., 2024) employ diffusion models to enhance the capability of capturing the visual details.

Our work is built on the latent generative world model due to its efficiency, but targets multi-agent settings by explicitly factorizing dynamics into agent and environment modules, enabling structured imagination of interactions and supporting decentralized execution.

**Model-based MARL.** Model-based MARL must handle partial observability of the environment and non-stationarity from concurrently adapting policies. Early approaches assume global observability (Krupnik et al., 2020; Zhang et al., 2021) or sufficiency of joint observations (Willemsen et al., 2021), sidestepping history dependencies. Later works pose learning multi-agent dynamics as a sequence modeling task using recurrent state-space model (Xu et al., 2022) or

transformer-based world models (Zhang et al., 2025; Liu et al., 2024b; Deihim et al., 2025). However, these methods focus on expressive trajectory modeling rather than enforcing a factorized latent structure that matches decentralized execution. Inspired by Dreamer (Hafner et al., 2020) and MuZero (Schrittwieser et al., 2020), latent world models have been adapted to MARL; however, many methods broadcast latent states among agents to maintain consistency across imagination and execution, thereby violating the CTDE paradigm. For instance, MAMBA (Egorov & Shpilman, 2022) leverages communication to maintain the consistency of agent's latent states for execution and imagination; Built on MAMBA, MAG (Wu et al., 2023) mitigates multi-step compounding errors in world modeling; CoDreamer (Toledo, 2024) employs GNN-based communication for both latent state updates and action selection; and MAZero (Liu et al., 2024a) utilizes a communication network to pool the latent states and actions for all agents to facilitate centralized planning. MABL (Venugopal et al., 2024) proposes a bi-level architecture to reduce communication requirements, but its global state transition model may not scale effectively as the number of agents increases.

Unlike previous methods that operate in the observation space or broadcast latent states, our work forces the latent states during imagination to reflect the factorized structure as in the real environment. This design leads to fully decentralized execution while enabling the world model to model the interactions between agents without the need for learning a global state transition model.

## 3. Preliminaries

**Learning latent dynamics.** Recurrent state-space models (RSSMs) (Hafner et al., 2019; 2020) learn action-conditioned generative dynamics that can roll out trajectories of observations and rewards given actions. An RSSM (parameterized by $\phi$) comprises: (1) a representation model $q_\phi(I_t \mid I_{t-1}, a_{t-1}, o_t)$[1] that infers the posterior latent state from the previous latent state $I_{t-1}$, the previous action $a_{t-1}$, and the current observation $o_t$; (2) a transition model $p_\phi(\hat{I}_t \mid I_{t-1}, a_{t-1})$ that predicts the next latent state from the previous state and action; and (3) decoders $p_\phi(\hat{o}_t \mid I_t)$ and $p_\phi(\hat{r}_t \mid I_t)$ that reconstruct the observation and reward from the latent state. These components are jointly trained to maximize the evidence lower bound (ELBO) on observed trajectories. This objective encourages high likelihood for the observed trajectories, while enforcing the consistency between the representation model and the transition model to capture the environment dynamics. During policy training, imagined trajectories are generated by first initializing

---

[1]We use the notation $I_t$ to emphasize it represents the internal state of an agent, which corresponds to $s_t$ in the original RSSM paper (Hafner et al., 2019).

$I_1$ from a subsequence of an observed trajectory sampled from the replay buffer, and then recursively predicting latent states $\hat{I}_{t+1} \sim p_\phi(\cdot \mid I_t, a_t)$ with action sampled from a learned policy $a_t \sim \pi_\theta(\cdot \mid I_t)$.

DreamerV2 (Hafner et al., 2021) further decomposes the latent state $I_t = (h_t, z_t)$ into a deterministic component $h_t$ and a stochastic component $z_t$. Given an observation $o_t$, the agent first updates the deterministic state via the recurrent model $h_t = f_\phi(I_{t-1}, a_{t-1})$, and then infers the stochastic state with the representation model $z_t \sim q_\phi(\cdot \mid h_t, o_t)$. During imagination, it reuses the recurrent model to update the deterministic state $h_t = f_\phi(I_{t-1}, a_{t-1})$, but samples the stochastic component from the transition predictor $\hat{z}_t \sim p_\phi(\cdot \mid h_t)$ without access to observations.

Our work builds on DreamerV2 to leverage its decomposition of the latent state into deterministic and stochastic components, where the deterministic components encode the local information of agents while the stochastic components capture the interaction between agents.

**Dec-POMDPs.** We consider cooperative, partially observable tasks formalized as decentralized partially observable Markov decision processes (Dec-POMDPs) (Oliehoek & Amato, 2016). In a Dec-POMDP with $n$ agents, each agent $i$ acts based on local information. At each timestep $t$, while the environment is in state $s_t$, agent $i$ receives a local observation $o_t^i \sim p(\cdot \mid s_t)$, which is appended to its local history $\tau_t^i = (o_{1:t}^i, a_{1:t-1}^i)$. Depending on the environment, agents may also receive an available action mask $m_t^i \in \{0,1\}^{|\mathcal{A}^i|}$ over discrete action space $\mathcal{A}^i$, indicating which actions are valid at timestep $t$; as well as a continuation flag $c_t^i \in \{0,1\}$ for finite-horizon tasks, where $c_t^i = 0$ indicates the end of an episode. Based on this history, the agent selects an action $a_t^i \sim \pi^i(\cdot \mid \tau_t^i)$, forming a joint action $a_t^{1:n} = (a_t^1, \ldots, a_t^n)$. Executing $a_t^{1:n}$ in state $s_t$ transitions the environment to $s_{t+1}$ and yields a shared reward $r_{t+1}$ according to $p(s_{t+1}, r_{t+1} \mid s_t, a_t^{1:n})$. Without loss of generality, we can extend this shared reward to agent-specific rewards $r_t^{1:n} = (r_t^1, \ldots, r_t^n)$, where for cooperative tasks, all agents share the same reward: $r_t^1 = \ldots = r_t^n = r_t$. The goal is to learn decentralized policies $\pi^{1:n} = (\pi^1, \ldots, \pi^n)$ that maximize the expected discounted return $\mathbb{E}_{\pi^{1:n}, p}[\sum_{t=1}^{\infty} \gamma^{t-1} r_{t+1}]$, where $\gamma \in (0,1)$ is the discount factor.

## 4. Method

We present DMAWM, a framework for learning decentralized policies in latent space. It features independent agent modules to maintain factorized latent states and a shared environment module to model interactions during imagination, effectively capturing multi-agent dynamics without sacrificing the decentralizability required for execution.

### 4.1. Framework

DMAWM comprises independent agent modules and a shared environment module, as shown in Figure 1. This design enables effective learning of disentangled latent states while capturing agent interactions. The agent modules independently process local observations to form factorized latent representations, while the environment module models agent interactions during imagination, generating imaginary trajectories for policy learning.

**Agent module.** Each agent module (parameterized by $\psi$) operates independently, maintaining its own internal state $I_t^i = (h_t^i, z_t^i)$ that consists of a deterministic component $h_t^i$ and a stochastic component $z_t^i$. Upon receiving a local observation $o_t^i$, the agent updates its state through two components: the recurrent model updates the deterministic part $h_t^i = f_\psi(I_{t-1}^i, a_{t-1}^i)$, while the representation model infers the stochastic part $z_t^i \sim q_\psi(\cdot \mid h_t^i, o_t^i)$, forming a factorized posterior $q_\psi(z_t^{1:n} \mid h_t^{1:n}, o_t^{1:n}) = \prod_{i=1}^n q_\psi(z_t^i \mid h_t^i, o_t^i)$, as the individual stochastic state is only conditioned on local information. Action selection is based solely on the agent's internal state, i.e., $a_t^i \sim \pi_\theta(\cdot \mid I_t^i)$, ensuring decentralized decision-making. Each agent module has the components:

$$
\begin{aligned}
\text{Recurrent model:} \quad & h_t^i = f_\psi(h_{t-1}^i, z_{t-1}^i, a_{t-1}^i) \\
\text{Representation model:} \quad & z_t^i \sim q_\psi(\cdot \mid h_t^i, o_t^i)
\end{aligned}
\tag{1}
$$

**Environment module.** The environment module (parameterized by $\phi$) plays a crucial role in generating imaginary trajectories for policy training. During imagination, it replaces the real environment to simulate the environment dynamics and inter-agent interactions. Given the previous joint latent state $I_{t-1}^{1:n}$ and actions $a_{t-1}^{1:n}$, the recurrent models of agent modules independently compute their deterministic components $h_t^i = f_\psi(I_{t-1}^i, a_{t-1}^i)$. Then the interaction predictor samples the joint stochastic components from the prior distribution $p_\phi(\hat{z}_t^{1:n} \mid h_t^{1:n})$, as the distribution does not condition on any real observations. Finally, dedicated decoders reconstruct trajectory components from the latent states, providing supervision signals for model learning.

The key innovation lies in the transformer-based interaction predictor which serves two purposes. (1) Modeling the dependencies between agents: During decentralized execution, agents' local observations contain information about the other agents. However, no observation is available in imagination. The interaction predictor fills the gap by explicitly modeling dependencies between agents. (2) Enforcing disentanglement to ensure decentralizability: As the interaction predictor has access to the privileged information of the joint deterministic state, it could introduce undesired correlations between the latent states of agents. We need to prevent this as agents do not have access to the

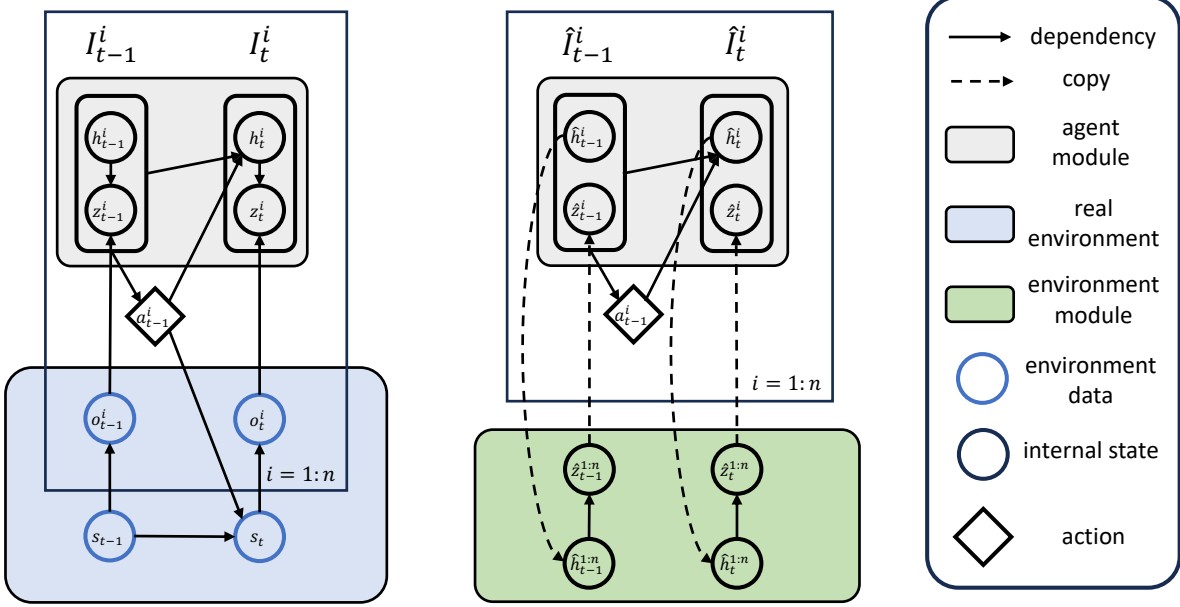

(a) Interaction with real environment     (b) Interaction with a shared environment module

*Figure 1.* Illustration of agent interactions. The gray area represents the agent module where the latent states are updated; the blue area represents the real environment which manages the state transition and observation generation; the green area represents the environment module which replaces the real environment during imagination and couples agents via their latent states.

joint deterministic state during decentralized execution. To achieve this, we align this joint prior $p_\phi(\hat{z}_t^{1:n} \mid h_t^{1:n})$ to the factorized posterior $\prod_{i=1}^{n} q_\psi(z_t^i \mid h_t^i, o_t^i)$ via the dynamics loss and representation loss (introduced in Section 4.2). The environment module has the components:

$$
\begin{aligned}
\text{Interaction predictor:} \quad & \hat{z}_t^{1:n} \sim p_\phi(\cdot \mid h_t^{1:n}) \\
\text{Observation decoder:} \quad & \hat{o}_t^i \sim p_\phi(\cdot \mid h_t^i, z_t^i) \\
\text{Reward decoder:} \quad & \hat{r}_t^{1:n} \sim p_\phi(\cdot \mid h_t^{1:n}) \quad (2) \\
\text{Continuation decoder:} \quad & \hat{c}_t^{1:n} \sim p_\phi(\cdot \mid h_t^{1:n}) \\
\text{Available actions decoder:} \quad & \hat{m}_t^{1:n} \sim p_\phi(\cdot \mid h_t^{1:n})
\end{aligned}
$$

### 4.2. Learning disentangled multi-agent world model

The learning objective of our multi-agent latent dynamics model is designed to maintain the factorized structure while effectively capturing agent interactions. Similar to Dreamer (Hafner et al., 2020), the training process iterates between collecting real environment data, learning the latent dynamics model, and training policies through imagination.

To train the multi-agent world model, we optimize the following objective:

$$
\mathcal{L}(\phi, \psi) = \underbrace{\beta_{\text{dyn}} \mathcal{L}_{\text{dyn}}(\phi, \psi)}_{\text{dynamics loss}} + \underbrace{\beta_{\text{rep}} \mathcal{L}_{\text{rep}}(\psi)}_{\text{representation loss}} + \underbrace{\beta_{\text{dec}} \mathcal{L}_{\text{dec}}(\phi, \psi)}_{\text{decoder loss}},
$$
$$(3)$$

where $\beta_{\text{dyn}}$, $\beta_{\text{rep}}$, and $\beta_{\text{dec}}$ are the weights for each loss term. The dynamics loss and representation loss align the prior and posterior distribution of the latent states, while the decoder loss encourages the model to reconstruct the real trajectories. Together, the above objective maximizes the variational lower bound of the likelihood of the real trajectories. The detailed derivation is in Appendix B.

**Dynamics loss and representation loss.** The dynamics and representation losses enforce disentanglement of latent states while still capturing their interactions during imagination. Since no observation is available during imagination, the interaction predictor takes the joint deterministic state as input to model the interaction between agents instead. By aligning the joint prior with the factorized posterior (illustrated in Figure 2), we compel the interaction predictor to generate a statistically consistent joint state, even in the absence of observations. We later verify this intended effect with a probing experiment in Section 5.4. We train the prior and the posterior via the following losses:

$$
\begin{aligned}
\mathcal{L}_{\text{dyn}}(\phi, \psi) &= \sum_{t=1}^{T} D_{\text{KL}}\left( \prod_{i=1}^{n} \text{sg}(q_\psi(\cdot \mid h_t^i, o_t^i)) \,\|\, p_\phi(\cdot \mid h_t^{1:n}) \right), \\
\mathcal{L}_{\text{rep}}(\psi) &= \sum_{t=1}^{T} D_{\text{KL}}\left( \prod_{i=1}^{n} q_\psi(\cdot \mid h_t^i, o_t^i) \,\|\, \text{sg}(p_\phi(\cdot \mid h_t^{1:n})) \right),
\end{aligned}
$$
$$(4)$$

where $\text{sg}(\cdot)$ denotes the stop-gradient operator and $T$ is the trajectory length. The dynamics loss $\mathcal{L}_{\text{dyn}}$ trains the

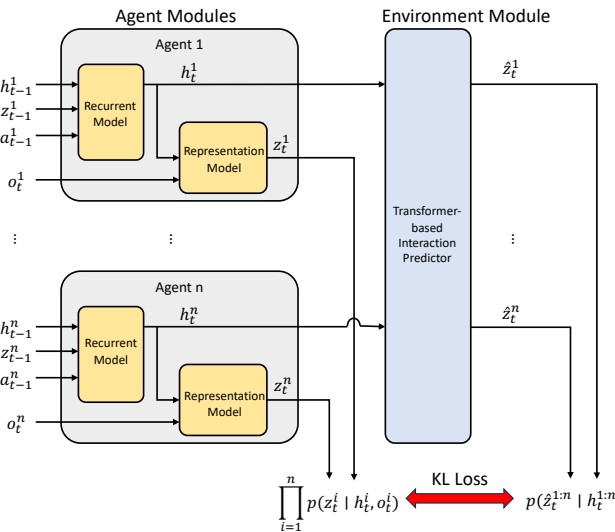

*Figure 2.* Illustration of the alignment of the joint prior and the factorized posterior. Agent modules form posterior latent states, while the environment module generates the joint prior latent states. They are aligned by the KL divergence to enforce disentanglement.

interaction predictor to match the factorized posterior, while the representation loss $\mathcal{L}_{\text{rep}}$ regularizes the posterior toward the joint prior. Using different weights for them allows us to use higher weight for optimizing the prior. $\mathcal{L}_{\text{dyn}}$ optimizes both $\phi$ and $\psi$ since $h_t^{1:n}$ is differentiable with respect to $\psi$.

**Decoder loss.** The decoders are trained to reconstruct the real trajectories, enabling the model to learn an informative representation of the environment and the structure of the underlying Dec-POMDP. The observation decoder reconstructs observations by taking both deterministic and stochastic states as input. Decoding the rewards, continuation flags, and available actions often require additional information beyond the local history, for example, an episode terminates when either all agents or all enemies are eliminated. To address this, we first combine all agents' deterministic states through a shared self-attention block, then decode the rewards, continuation flags, and available actions using separate heads. The decoder loss comprises:

$$\mathcal{L}_{\text{dec}}(\phi, \psi) = -\sum_{t=1}^{T} \left( \sum_{i=1}^{n} \log p_\phi(o_t^i \mid h_t^i, z_t^i) \right.$$
$$\left. + \log p_\phi(r_t^{1:n}, c_t^{1:n}, m_t^{1:n} \mid h_t^{1:n}) \right), \quad (5)$$

where $\{o_t^{1:n}, r_t^{1:n}, c_t^{1:n}, m_t^{1:n}\}_{t=1}^{T}$ are the ground-truth trajectory components. $\mathcal{L}_{\text{dec}}$ optimizes both $\phi$ and $\psi$ as $h_t^{1:n}$ is differentiable with respect to $\psi$. The loss in Equation 3 is optimized end-to-end with backpropagation through time (BPTT), with the gradient of discrete latent states estimated by the straight-through estimator (Bengio et al., 2013).

## 4.3. Learning decentralized policies in latent space

The disentangled latent states learned by our model facilitate the effective training of decentralized policies via imagination. The multi-agent dynamics within DMAWM naturally form a factored $n$-agent Dec-MDP (Bernstein et al., 2002), where the joint latent state $\hat{I}_t^{1:n} = (\hat{h}_t^i, \hat{z}_t^i)_{i=1}^{n}$ uniquely determines the underlying state (see Appendix A).

**Generating multi-agent trajectories in imagination.** Imaginary rollouts are initialized by encoding a subsequence of a ground-truth trajectory $\tau_H^{1:n} = (o_{1:H}^{1:n}, a_{1:H-1}^{1:n})$ using the agent module, where $H$ represents the subsequence length. The resulting joint latent state $(\hat{h}_1^{1:n}, \hat{z}_1^{1:n})$ serves as the starting point for imagination. At each imagined step $t$, the decoders first predict the reward $\hat{r}_t^{1:n}$, available action mask $\hat{m}_t^{1:n}$ and continuation flag $\hat{c}_t^{1:n}$ from $\hat{h}_t^{1:n}$. For each agent $i$, the actor model then samples a valid action $a_t^i \sim \pi_\theta(\cdot \mid \hat{h}_t^i, \hat{z}_t^i)$ by masking unavailable actions with $\hat{m}_t^i$, and the recurrent model updates the deterministic state $\hat{h}_{t+1}^i = f_\psi(\hat{h}_t^i, \hat{z}_t^i, a_t^i)$. The interaction predictor then samples the joint stochastic state $\hat{z}_{t+1}^{1:n} \sim p_\phi(\cdot \mid \hat{h}_{t+1}^{1:n})$ for all agents. This process repeats for $L$ steps to produce an imaginary trajectory.

**On-policy critic loss.** To accurately estimate the value, the centralized critic $v_\xi^{1:n}(\hat{h}_t^{1:n})$ utilizes a transformer to contextualize the joint deterministic state and estimates the value for each agent. The on-policy objective is optimized using the imaginary trajectory, which is sampled by the current policy. We bootstrap the value at the next step and calculate the advantage using the Generalized Advantage Estimator (GAE) (Schulman et al., 2021):

$$\delta_t^i = \hat{r}_t^i + \gamma \hat{c}_{t+1}^i v_\xi^i(\hat{h}_{t+1}^{1:n}) - v_\xi^i(\hat{h}_t^{1:n}),$$
$$A_t^i = \delta_t^i + \lambda \gamma \hat{c}_{t+1}^i A_{t+1}^i,$$

where $\lambda$ governs the bias-variance trade-off. The on-policy critic loss is computed over the imaginary trajectory:

$$\mathcal{L}_{\text{critic}}^{\text{on}}(\xi) = \sum_{t=1}^{L} \sum_{i=1}^{n} \left( \text{sg}(v_\xi^i(\hat{h}_t^{1:n}) + A_t^i) - v_\xi^i(\hat{h}_t^{1:n}) \right)^2. \quad (6)$$

**Off-policy critic loss.** In contrast to Dreamer, which trains exclusively on imaginary trajectories, we also ground the critic using real trajectories to ensure value estimates are consistent with ground-truth returns. To mitigate variance in importance sampling ratios, we employ truncated importance sampling (TIS) (Ionides, 2008):

$$\mathcal{L}_{\text{critic}}^{\text{off}}(\xi) = \sum_{t=1}^{T} \sum_{i=1}^{n} \min(\rho_{t:T}, C)(v_\xi^i(h_t^{1:n}) - V_t^i)^2, \quad (7)$$

where $\rho_{t:T} = \prod_{k=t}^{T} \prod_{i=1}^{n} \frac{\pi_\theta(a_k^i \mid h_k^i, z_k^i)}{\pi_\beta(a_k^i \mid h_k^i, z_k^i)}$ denotes the cumulative importance weight, $C$ is the truncation threshold, $V_t^i$ is

the return calculated from real trajectories, $h_t^{1:n}, z_t^{1:n}$ denote latent states encoded from real trajectories, and $\pi_\beta$ denotes the original policy for data collection.

The combined critic loss is given by:

$$\mathcal{L}_{\text{critic}}(\xi) = \mathcal{L}_{\text{critic}}^{\text{on}}(\xi) + \beta_{\text{off}}\mathcal{L}_{\text{critic}}^{\text{off}}(\xi), \qquad (8)$$

where $\beta_{\text{off}}$ is the coefficient for the off-policy loss term.

**Actor loss.** We train the actor using the PPO objective (Schulman et al., 2017) on imaginary trajectories. The actor loss is computed using the advantages $A_t^i$ estimated by the critic:

$$\mathcal{L}_{\text{actor}}(\theta) = -\sum_{t=1}^{L}\sum_{i=1}^{n}\Big[\min\big(\rho_t^i A_t^i, \text{clip}(\rho_t^i, 1-\epsilon, 1+\epsilon)A_t^i\big) \\ + \beta_{\text{ent}}\mathcal{H}(\pi_\theta(\cdot \mid \hat{h}_t^i, \hat{z}_t^i))\Big],$$

$$(9)$$

where $\rho_t^i = \frac{\pi_\theta(a_t^i|\hat{h}_t^i,\hat{z}_t^i)}{\pi_{\theta^{\text{old}}}(a_t^i|\hat{h}_t^i,\hat{z}_t^i)}$ is the probability ratio between the current and previous policies, the clipping operation constrains this ratio to $[1-\epsilon, 1+\epsilon]$, $\mathcal{H}(\cdot)$ denotes the policy entropy, and $\beta_{\text{ent}}$ is the entropy coefficient.

# 5. Experiments

We evaluate DMAWM on three challenging MARL benchmarks, comparing it against both model-based and model-free baselines. Beyond performance comparison, we also measure the runtime efficiency (Section 5.2), ablate the main components to study their individual contributions (Section 5.3), probe the learned latent states to verify disentanglement (Section 5.4), and visualize generated imaginary trajectories (Section 5.5).

## 5.1. Experimental setup

**Benchmarks.** We evaluate our method on three MARL benchmarks that pose complementary challenges: SMAC (Samvelyan et al., 2019) features complex dynamics and diverse coordination patterns; SMACv2 (Ellis et al., 2023), an updated version of SMAC, introduces greater randomness in the starting positions and unit types; Melting Pot (Leibo et al., 2021) uses visual observations and requires behavior switching according to the context. We train all algorithms for 400K environment steps on each benchmark. A detailed description of the benchmarks is provided in Appendix E.

**Baselines.** We compare our method against both model-based and model-free baselines. The model-based baselines include MBVD (Xu et al., 2022), which reconstructs observations to train a value decomposition method; MAMBA (Egorov & Shpilman, 2022), a current SOTA

method based on DreamerV2; and MARIE (Zhang et al., 2025), a recent transformer-based multi-agent world model method. The model-free baselines include QMIX (Rashid et al., 2018), MAPPO (Yu et al., 2022), and MAT (Wen et al., 2022).

**Implementation details.** We perform one training step of both the world model and the policies every 32 environment steps, beginning after an initial 5000-step warm-up. The world model is trained on batches of 16 sequences, each with length 64, sampled from a replay buffer of size $2.5 \times 10^5$. To handle agents that die or leave the scene, we introduce an absorbing latent state: absent agents observe a fixed null observation, execute a fixed no-operation action, and remain in this state for the rest of the episode. During imagination, the learned model generates imaginary trajectories of 4 steps with 1024 parallel rollouts. For Melting Pot, which uses visual observations, inputs are downsampled from $88 \times 88$ to $44 \times 44$ pixels to reduce GPU memory usage. Visual observations are encoded with a CNN and decoded with a transposed CNN (Dumoulin & Visin, 2016). To improve sample efficiency, we share parameters of the agent modules and policies across all agents. To ensure fairness for comparison, all algorithms use the same set of hyperparameters across benchmarks. Additional implementation details and hyperparameters are provided in Appendix C and Appendix D, respectively.

## 5.2. Main results

**Performance comparison.** Table 1 and Figure 3 summarize the results. We observe that DMAWM is consistently sample-efficient, matching or surpassing strong baselines in most tasks considered. On SMAC, DMAWM demonstrates superior learning speed and attains the highest win rates, particularly on the 2c_vs_64zg and corridor maps. In the more stochastic SMACv2 environment, both DMAWM and MAMBA outperform other baselines. We note that DMAWM exhibits substantially faster convergence and achieves the highest win rates, surpassing MAPPO even when the latter is trained for 10 times more environment steps. For the vision-based scenarios from Melting Pot, DMAWM obtains the highest returns without task-specific tuning, highlighting its effectiveness in modeling agent interactions within the visual domain. Table 8 further compares DMAWM with MARIE (Zhang et al., 2025), a closely related transformer-based multi-agent world model, on the benchmark used in the original MARIE paper. Under the same experimental protocol, DMAWM matches or outperforms MARIE on 10 out of 13 maps, with particularly large gains on harder SMAC maps such as 3s_vs_5z and 2c_vs_64zg. In Appendix I, we show the training results up to 1M steps, demonstrating that DMAWM continues to improve with extended training.

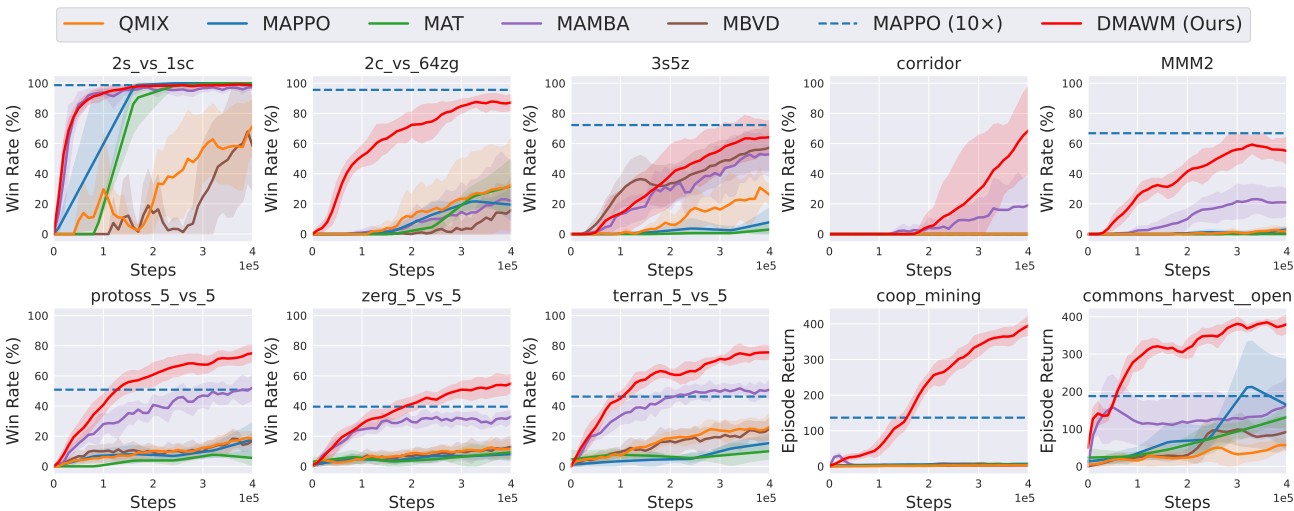

*Figure 3.* Training curves comparing DMAWM with model-based and model-free baselines on three MARL benchmarks: SMAC, SMACv2, and Melting Pot. Results are averaged over 5 independent runs, with shaded areas representing the standard deviation. All algorithms are trained for 400K environment steps. Dashed lines show performance of MAPPO after 10 times more environment steps than model-based algorithms.

*Table 1.* Performance comparison across SMAC, SMACv2, and Melting Pot benchmarks. We compared our approach against both model-based and model-free baselines. Evaluation metrics are win rate (%) for SMAC and SMACv2, and episode return for Melting Pot. All results are reported as the average over 5 independent runs, accompanied by their standard deviations.

| Benchmarks | Maps | Model-free | | | Model-based | | |
|---|---|---|---|---|---|---|---|
| | | QMIX | MAPPO | MAT | MAMBA | MBVD | DMAWM (Ours) |
| SMAC | 2s_vs_1sc | 69.8 (19.8) | **100.0** (0.0) | **100.0** (0.0) | 96.9 (2.3) | 62.1 (30.3) | 98.3 (1.4) |
| | 2c_vs_64zg | 31.8 (26.1) | 19.8 (7.7) | 30.9 (14.3) | 22.8 (25.0) | 14.8 (18.3) | **86.5** (4.7) |
| | 3s5z | 28.9 (8.8) | 7.2 (7.4) | 2.7 (3.5) | 53.2 (12.3) | 59.2 (16.1) | **65.3** (7.6) |
| | corridor | 0.0 (0.0) | 0.0 (0.0) | 0.0 (0.0) | 18.6 (17.7) | 0.0 (0.0) | **71.5** (25.9) |
| | MMM2 | 2.3 (2.3) | 2.7 (1.7) | 0.0 (0.0) | 21.2 (8.1) | 0.2 (0.7) | **54.7** (9.3) |
| SMACv2 | protoss_5_vs_5 | 19.0 (4.0) | 16.9 (9.2) | 5.9 (3.8) | 51.4 (8.3) | 14.9 (6.3) | **77.0** (5.8) |
| | zerg_5_vs_5 | 11.3 (5.9) | 7.9 (3.6) | 9.0 (3.8) | 32.4 (4.0) | 12.7 (6.5) | **55.3** (5.0) |
| | terran_5_vs_5 | 25.6 (8.4) | 15.0 (4.9) | 9.8 (4.7) | 50.9 (5.3) | 24.1 (7.7) | **76.0** (3.8) |
| Melting Pot | Coop Mining | 2.8 (2.4) | 6.6 (2.7) | 4.9 (1.0) | 2.6 (2.1) | 5.7 (2.6) | **405.7** (23.6) |
| | Commons Harvest: Open | 61.5 (13.4) | 170.6 (105.2) | 127.6 (23.5) | 175.9 (69.4) | 105.0 (30.1) | **375.9** (23.3) |

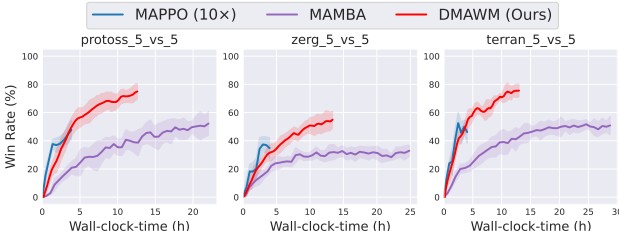

*Figure 4.* Wall-clock-time comparison between DMAWM, MAPPO, and MAMBA on the SMACv2 benchmark. Results are averaged over 3 independent runs. DMAWM and MAMBA are trained for 400K environment steps, while MAPPO is trained for 4M environment steps.

**Wall-clock-time comparison.** To evaluate the runtime efficiency of DMAWM, we compare the wall-clock-time of DMAWM against MAPPO and MAMBA. All methods are

benchmarked under a single NVIDIA RTX 3090 GPU. As shown in Figure 4, DMAWM achieves comparable runtime efficiency as MAPPO while using far fewer environment steps to reach the same performance, narrowing the runtime gap between model-based and model-free algorithms. More results can be found in Appendix F.

### 5.3. Ablation study

To assess the individual contributions of DMAWM's core components—the transformer-based critic, interaction predictor, and off-policy critic loss—we conduct ablation studies on them. To ablate the transformer-based critic, we replace it with an MLP that estimates the value based on the latent states of an agent. We call this ablation DMAWM-TC. The second ablation, DMAWM-IP, replaces the interaction predictor with an MLP, wherein each agent independently

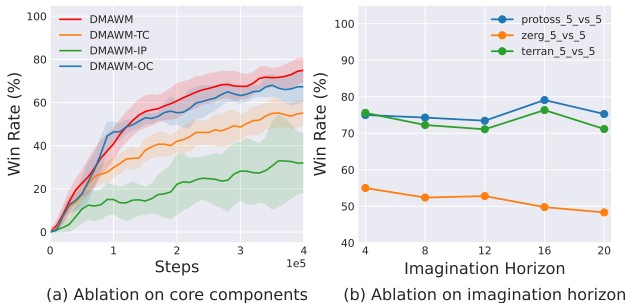

(a) Ablation on core components    (b) Ablation on imagination horizon

*Figure 5.* Ablation study on SMACv2, all results are averaged across the 3 independent runs. (a) We remove each of the core components to study the individual contributions on the protoss_5_vs_5 map. (b) We study the impact of the imagination horizon on the performance of DMAWM. Each map is trained for 400K environment steps.

predicts its own latent state and other trajectory components. The third ablation, DMAWM-OC, removes the off-policy critic loss by setting $\beta_{\text{off}} = 0$, training the critic exclusively on imaginary trajectories. As shown in Figure 5 (a), all ablations perform worse than the full DMAWM, underscoring the importance of these components.

In Figure 5 (b), we also study the impact of the imagination horizon on the performance of DMAWM. The results show that DMAWM remains stable across the tested horizons, but performance does not further improve with longer rollouts.

### 5.4. Probing latent state disentanglement

When the world model is entangled, an agent's action may significantly impact the latent states of other agents, even if they have little interaction with each other. This is caused by the information leakage, where local information is propagated to the other agents' latent states via the interaction modeling process. To investigate whether different methods suffer from this issue, we design a probing experiment to detect whether the information leakage occurs.

**Setup.** We conduct the probing experiment on the protoss_5_vs_5 map of SMACv2, comparing DMAWM with MAMBA. MAMBA (Egorov & Shpilman, 2022) is a representative method that leverages communication to maintain the consistency of agent's latent states for execution and imagination. For each algorithm, we first train the world model for 400K steps using data collected by random policies, and then freeze its parameters. We then train action predictors $p_\theta(a_t^{-i} \mid \ell_t^i, \hat{\ell}_{t+1}^i)$ to infer other agents' actions $a_t^{-i} = (a_t^1, \ldots, a_t^{i-1}, a_t^{i+1}, \ldots, a_t^n)$ from agent $i$'s latent transition distributions, where $\ell_t^i$ denotes the posterior logits of $q_\psi(z_t^i \mid h_t^i, o_t^i)$ inferred from the real trajectory, and $\hat{\ell}_{t+1}^i$ denotes the prior logits of $p_\phi(\hat{z}_{t+1}^i \mid h_{t+1}^{1:n})$ predicted by the world model. As a baseline, we train an observation-based action predictor $p_\theta(a_t^{-i} \mid o_t^i, o_{t+1}^i)$ using only agent $i$'s local

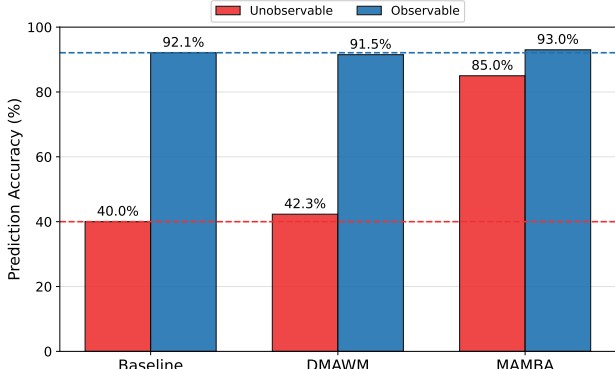

*Figure 6.* Action prediction accuracy on the protoss_5_vs_5 map. "Observable" refers to teammates within field of view; "Unobservable" refers to teammates outside field of view. The observation-based predictor serves as a baseline for expected accuracy without information leakage.

observations. We train all action predictors for another 200K environment steps and report prediction accuracy separately for observable and unobservable teammates.

**Results.** The results are reported in Figure 6. For observable teammates, both predictors achieve high accuracy, validating that the latent states capture relevant information about visible agents. The critical difference comes from unobservable teammates: DMAWM's predictor (42.3%) performs comparably to the observation-based baseline (40.0%). In contrast, MAMBA's predictor achieves 85.0% accuracy, revealing that significant global information leaks into individual agent states. This confirms that DMAWM successfully maintains disentangled representations suitable for decentralized execution.

### 5.5. Visualization of the latent space

To qualitatively evaluate the multi-agent latent dynamics model's ability to capture agent interactions, we generate an imaginary trajectory in the Coop Mining environment and compare it with the ground-truth trajectory, as shown in Figure 7. The figure displays the decoded observations of two out of the four agents within the same trajectory. We use the representation model to encode the initial 6 frames and generate the subsequent 24 frames with the interaction predictor and the observation decoder. From the visualization, we can see that the positions of agents and the structure of the wall in the imagination align well with the real environment. While the ore distribution in imagination (iron ore is marked in gray and gold ore is marked in yellow) aligns more closely with the ground-truth trajectory at early stage, they diverge at later timesteps. This is expected since the models must infer ore locations in unobserved areas. What we find interesting is that the relative positions of the two agents in imagination remain

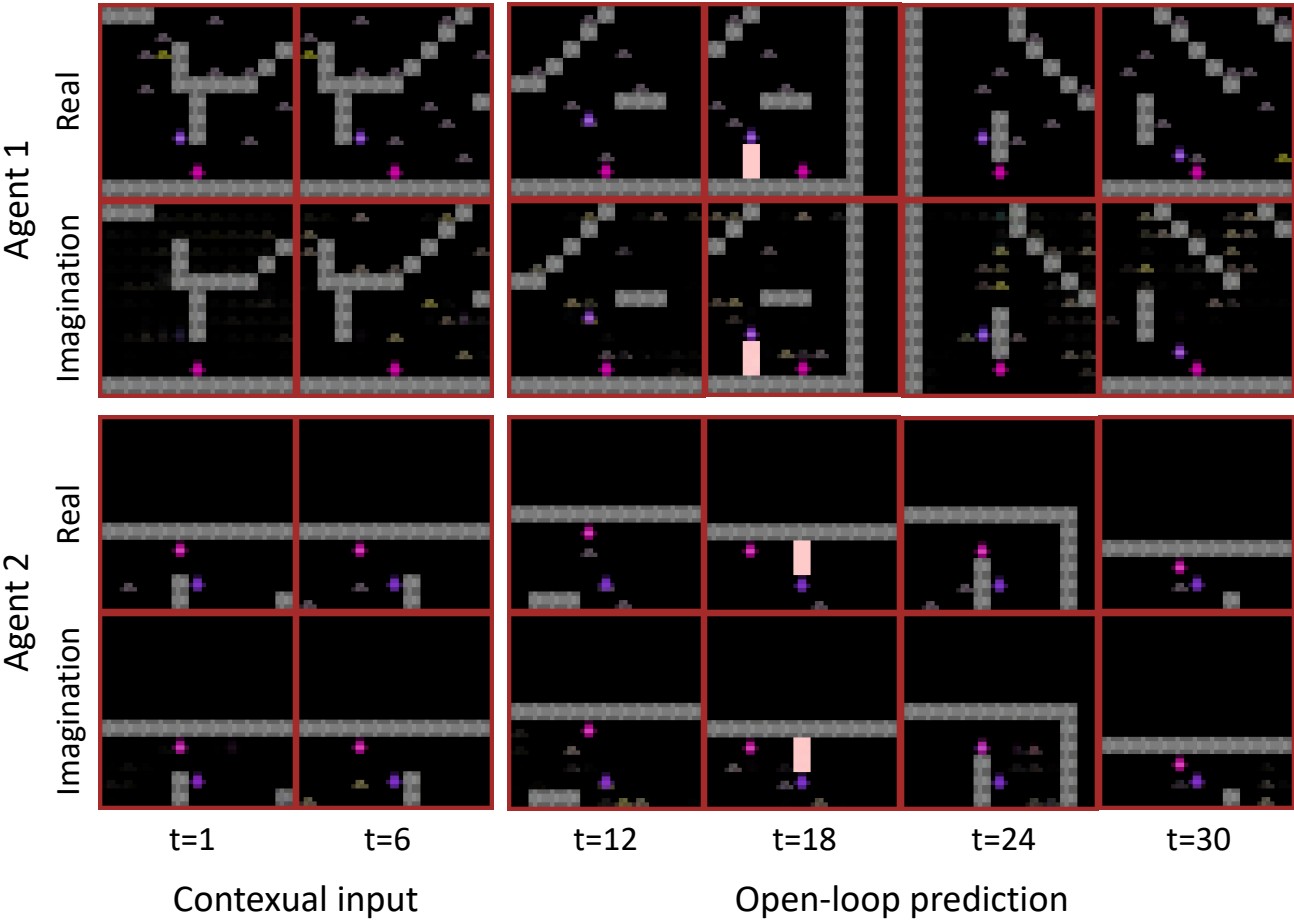

*Figure 7.* Long-horizon trajectory prediction by our multi-agent latent dynamics model in the Coop Mining environment. Conditioned on an initial 6-frame context from a hold-out trajectory and corresponding actions, the model employs the interaction predictor to generate 24 subsequent frames in the latent space. We selectively show the decoded observations for two out of the four agents, to demonstrate the model's ability to capture multi-agent interactions coherently over long horizons.

consistent with each other in their ego-centric observations throughout all timesteps. Notably, at $t = 18$, agent 2's action (mining beam) is accurately reflected in agent 1's decoded observation, highlighting the model's efficacy in capturing agent interactions.

## 6. Conclusion and Future Work

In this work, we addressed the challenge of learning decentralized policies for multi-agent tasks using world models. We introduced the Disentangled Multi-Agent World Model (DMAWM), a framework that learns decentralized policies in the latent space with a novel architecture featuring independent agent modules and a shared environment module. This architecture enables it to learn a factorized latent representation that explicitly captures agent interactions while effectively disentangling individual agent latent states. This disentanglement is crucial for training decentralized policies via imagined trajectories, and we designed a probing experi-

ment to verify that the learned latent representation indeed disentangles individual agent states. Our experiments on challenging MARL benchmarks, with both vector and visual observations, demonstrated that DMAWM significantly outperforms existing model-based and model-free baselines in both sample efficiency and final performance.

While this paper mainly focuses on cooperative tasks, extending DMAWM to mixed-motive scenarios is a promising direction, as the core mechanisms are not inherently tied to cooperative tasks. Another important future direction is improving scalability to larger populations by exploiting sparse attention or networked-MDP topology, allowing the world model to focus on interactions most relevant for prediction and control. We also find that multi-task multi-agent world models are worth investigating. Although tasks may vary, the local dynamics of agents are likely to be similar. This implies the potential to learn a more generalizable world model capable of adapting to different scenarios without the need for task-specific training.

## Acknowledgements

We thank the annonymous reviewers for their valuable feedback and suggestions. This work was supported by the National Natural Science Foundation of China (No. 62276126, 62495090, and 62495094), the "111 Center" (No. B26023), and the Fundamental and Interdisciplinary Disciplines Breakthrough Plan of the Ministry of Education of China (No. JYB2025XDXM118).

## Impact Statement

This paper presents work whose goal is to advance the field of Machine Learning. There are many potential societal consequences of our work, none of which we feel must be specifically highlighted here.

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

## A. Connections between multi-agent latent dynamics and traditional multi-agent formulations

In this section, we build the connections between the multi-agent latent dynamics and traditional multi-agent formulations, hopefully the techniques developed for these specific domains can be applied to our framework.

The multi-agent latent dynamics model can be seen as an action-conditioned generative model of the joint observation sequence that can be factorized as:

$$
\begin{aligned}
p(o_{1:T}^{1:n}, I_{1:T}^{1:n} \mid a_{1:T-1}^{1:n}) &= p(I_{1:T}^{1:n} \mid a_{1:T-1}^{1:n}) p(o_{1:T}^{1:n} \mid I_{1:T}^{1:n}) \\
&= \prod_{t=1}^{T} p(I_t^{1:n} \mid I_{t-1}^{1:n}, a_{t-1}^{1:n}) p(o_t^{1:n} \mid I_t^{1:n}) \\
&= \prod_{t=1}^{T} p(I_t^{1:n} \mid I_{t-1}^{1:n}, a_{t-1}^{1:n}) \prod_{i=1}^{n} p(o_t^i \mid I_t^i),
\end{aligned}
\tag{10}
$$

where $p(I_t^{1:n} \mid I_{t-1}^{1:n}, a_{t-1}^{1:n})$ is the transition probability and $p(o_t^i \mid I_t^i)$ is the observation probability.

To formulate the interaction derived from the above multi-agent latent dynamics model, we can define the state as the collection of the individual latent states of all agents $s_t = (I_t^1, \ldots, I_t^n)$ as the sequence $(I_t^{1:n})_{t=1}^T$ is Markovian on each of its components. The transition probability is governed by the multi-agent latent dynamics model $p(s_t \mid s_{t-1}, a_{t-1}^{1:n}) = p(I_t^{1:n} \mid I_{t-1}^{1:n}, a_{t-1}^{1:n})$. Each agent could observe its component of the joint latent state $o_t^i = I_t^i$, leading to the fact that the joint observation is equivalent to the state, i.e., $o_t^{1:n} = I_t^{1:n} = s_t$. This interaction formulation is captured by an *agent-wise factored Dec-MDP* (Goldman & Zilberstein, 2004), where the state is uniquely determined by the joint observation of all agents.

## B. Derivation of the ELBO

Here we derive the evidence lower bound (ELBO) for the joint observation sequence $o_{1:T}^{1:n}$ given the action sequence $a_{1:T-1}^{1:n}$. The derivations for the other trajectory components are similar.

The posterior is the representation model that updates recursively:

$$
q(I_{1:T}^{1:n} \mid a_{1:T-1}^{1:n}, o_{1:T}^{1:n}) = \prod_{t=1}^{T} q(I_t^{1:n} \mid I_{t-1}^{1:n}, a_{t-1}^{1:n}, o_t^{1:n}).
\tag{11}
$$

We also need the marginal posterior to derive the ELBO:

$$
\begin{aligned}
\sum_{I_{t \neq l}^{1:n}} q(I_{1:T}^{1:n} \mid a_{1:T-1}^{1:n}, o_{1:T}^{1:n}) &= \sum_{I_{1:l-1}^{1:n}} \sum_{I_{l+1:T}^{1:n}} q(I_{1:T}^{1:n} \mid a_{1:T-1}^{1:n}, o_{1:T}^{1:n}) \\
&= \sum_{I_{1:l-1}^{1:n}} \sum_{I_{l+1:T}^{1:n}} \prod_{t=1}^{T} q(I_t^{1:n} \mid I_{t-1}^{1:n}, a_{t-1}^{1:n}, o_t^{1:n}) \qquad \text{Apply Eq. 11} \\
&= \sum_{I_{1:l-1}^{1:n}} \sum_{I_{l+1:T}^{1:n}} \prod_{t=1}^{l} q(I_t^{1:n} \mid I_{t-1}^{1:n}, a_{t-1}^{1:n}, o_t^{1:n}) \prod_{t=l+1}^{T} q(I_t^{1:n} \mid I_{t-1}^{1:n}, a_{t-1}^{1:n}, o_t^{1:n}) \\
&= \sum_{I_{1:l-1}^{1:n}} \underbrace{\prod_{t=1}^{l} q(I_t^{1:n} \mid I_{t-1}^{1:n}, a_{t-1}^{1:n}, o_t^{1:n})}_{=q(I_{1:l}^{1:n} \mid a_{1:l-1}^{1:n}, o_{1:l}^{1:n}) \quad \text{by Eq. 11}} \underbrace{\sum_{I_{l+1:T}^{1:n}} \prod_{t=l+1}^{T} q(I_t^{1:n} \mid I_{t-1}^{1:n}, a_{t-1}^{1:n}, o_t^{1:n})}_{=1} \\
&= \sum_{I_{1:l-1}^{1:n}} q(I_{1:l}^{1:n} \mid a_{1:l-1}^{1:n}, o_{1:l}^{1:n}) = q(I_l^{1:n} \mid a_{1:l-1}^{1:n}, o_{1:l}^{1:n}).
\end{aligned}
\tag{12}
$$

The ELBO is derived as follows:

$$\ln p(o_{1:T}^{1:n} \mid a_{1:T-1}^{1:n})$$

$$= \ln \sum_{I_{1:T}^{1:n}} p(o_{1:T}^{1:n}, I_{1:T}^{1:n} \mid a_{1:T-1}^{1:n})$$

$$= \ln \sum_{I_{1:T}^{1:n}} p(I_{1:T}^{1:n} \mid a_{1:T-1}^{1:n}) p(o_{1:T}^{1:n} \mid I_{1:T}^{1:n}) \qquad \text{Apply Eq. 10}$$

$$= \ln \sum_{I_{1:T}^{1:n}} q(I_{1:T}^{1:n} \mid a_{1:T-1}^{1:n}, o_{1:T}^{1:n}) \frac{p(I_{1:T}^{1:n} \mid a_{1:T-1}^{1:n})}{q(I_{1:T}^{1:n} \mid a_{1:T-1}^{1:n}, o_{1:T}^{1:n})} p(o_{1:T}^{1:n} \mid I_{1:T}^{1:n})$$

$$= \ln \mathbb{E}_{q(I_{1:T}^{1:n} \mid a_{1:T-1}^{1:n}, o_{1:T}^{1:n})} \left[ \frac{p(I_{1:T}^{1:n} \mid a_{1:T-1}^{1:n})}{q(I_{1:T}^{1:n} \mid a_{1:T-1}^{1:n}, o_{1:T}^{1:n})} p(o_{1:T}^{1:n} \mid I_{1:T}^{1:n}) \right]$$

$$\geq \mathbb{E}_{q(I_{1:T}^{1:n} \mid a_{1:T-1}^{1:n}, o_{1:T}^{1:n})} \left[ \ln \frac{p(I_{1:T}^{1:n} \mid a_{1:T-1}^{1:n})}{q(I_{1:T}^{1:n} \mid a_{1:T-1}^{1:n}, o_{1:T}^{1:n})} p(o_{1:T}^{1:n} \mid I_{1:T}^{1:n}) \right] \qquad \text{Jensen's inequality}$$

$$= \mathbb{E}_{q(I_{1:T}^{1:n} \mid a_{1:T-1}^{1:n}, o_{1:T}^{1:n})} \left[ \sum_{t=1}^{T} \ln p(o_t^{1:n} \mid I_t^{1:n}) - \sum_{t=1}^{T} \ln \frac{q(I_t^{1:n} \mid I_{t-1}^{1:n}, a_{t-1}^{1:n}, o_t^{1:n})}{p(I_t^{1:n} \mid I_{t-1}^{1:n}, a_{t-1}^{1:n})} \right]$$

$$= \sum_{t=1}^{T} \mathbb{E}_{q(I_{1:T}^{1:n} \mid a_{1:T-1}^{1:n}, o_{1:T}^{1:n})} \left[ \ln p(o_t^{1:n} \mid I_t^{1:n}) \right] - \sum_{t=1}^{T} \mathbb{E}_{q(I_{1:T}^{1:n} \mid a_{1:T-1}^{1:n}, o_{1:T}^{1:n})} \left[ \ln \frac{q(I_t^{1:n} \mid I_{t-1}^{1:n}, a_{t-1}^{1:n}, o_t^{1:n})}{p(I_t^{1:n} \mid I_{t-1}^{1:n}, a_{t-1}^{1:n})} \right]$$

$$= \sum_{t=1}^{T} \underbrace{\mathbb{E}_{q(I_{1:t}^{1:n} \mid o_{1:t}^{1:n}, a_{1:t-1}^{1:n})}}_{\text{marginalized by Eq. 12}} \left[ \ln p(o_t^{1:n} \mid I_t^{1:n}) \right]$$

$$\quad - \sum_{t=1}^{T} \underbrace{\mathbb{E}_{q(I_{t-1}^{1:n} \mid o_{1:t-1}^{1:n}, a_{1:t-2}^{1:n})}}_{\text{marginalized by Eq. 12}} \left[ D_{\text{KL}} \left( q(\cdot \mid I_{t-1}^{1:n}, a_{t-1}^{1:n}, o_t^{1:n}) \,\|\, p(\cdot \mid I_{t-1}^{1:n}, a_{t-1}^{1:n}) \right) \right]$$

$$= \sum_{t=1}^{T} \mathbb{E}_{q(I_t^{1:n} \mid o_{1:t}^{1:n}, a_{1:t-1}^{1:n})} \left[ \sum_{i=1}^{n} \ln p(o_t^i \mid I_t^i) \right]$$

$$\quad - \sum_{t=1}^{T} \mathbb{E}_{q(I_{t-1}^{1:n} \mid o_{1:t-1}^{1:n}, a_{1:t-2}^{1:n})} \left[ D_{\text{KL}} \left( \prod_{i=1}^{n} q(\cdot \mid I_{t-1}^i, a_{t-1}^i, o_t^i) \,\|\, p(\cdot \mid I_{t-1}^{1:n}, a_{t-1}^{1:n}) \right) \right].$$

## C. Other implementation details

In multi-agent tasks, agents can become absent due to death or leaving the scene (Samvelyan et al., 2019; Li et al., 2022). Drawing inspiration from prior work (Schrittwieser et al., 2020; Egorov & Shpilman, 2022), we address this by incorporating an absorbing state into our latent dynamics model to represent agent absence. When an agent is absent, it immediately transitions to the absorbing state and remains there indefinitely, where the agent constantly observes a fixed null observation and executes a fixed non-operation action. When an episode is terminated in the environment, we let all agents remain in the absorbing state in the latent space but the received rewards are zero. To facilitate the model's learning of this behavior, we (1) relabel the observations, available actions, and continuation flags of absent agents within an episode, and (2) append null observations, available actions, zero rewards, and zero continuation flags to the end of the trajectory. This approach enables the latent dynamics model to effectively learn and represent agent absence.

The interaction predictor is implemented using a Transformer network (Vaswani et al., 2017). It is tasked with predicting the discrete latent state, conditioned on the joint deterministic state $p_\phi(z_t^{1:n} \mid h_t^{1:n})$. Each discrete latent state is represented by 32 one-hot vectors, each with 32 classes. The prediction process begins by encoding the joint deterministic state $h_t^{1:n}$ using a Transformer encoder, which yields $(\bar{h}_t^1, \ldots, \bar{h}_t^n) = \text{TransformerEncoder}(h_t^1, \ldots, h_t^n)$. Subsequently, a Multi-Layer Perceptron (MLP) maps each resulting embedding $\bar{h}_t^i$ to a 1024-dimensional vector. This vector is then reshaped to facilitate the sampling of the discrete latent state.

We also considered an autoregressive interaction predictor that factorizes the prior as $\prod_{i=1}^{n} p_\phi(z_t^i \mid h_t^{1:n}, z_t^{<i})$ and samples the agents' stochastic states sequentially. Although this variant is more expressive, we found it substantially less stable during training. We attribute this to two practical issues: first, the autoregressive ordering introduces an arbitrary dependency among agents and weakens the permutation symmetry of the interaction model; second, sequential sampling compounds stochastic variation across agents, making imagined trajectories noisier. We therefore use the parallel interaction predictor, which captures inter-agent dependencies through the joint deterministic context $h_t^{1:n}$ while preserving stable parallel sampling of per-agent stochastic states.

The reward decoder $p_\phi(\hat{r}_t^{1:n} \mid h_t^{1:n})$, continuation decoder $p_\phi(\hat{c}_t^{1:n} \mid h_t^{1:n})$, and available actions decoder $p_\phi(\hat{m}_t^{1:n} \mid h_t^{1:n})$ are implemented with a shared Transformer network. First, the shared Transformer encoder blocks process the joint deterministic state $h_t^{1:n}$, producing $(\tilde{h}_t^1, \ldots, \tilde{h}_t^n) = \text{TransformerEncoder}(h_t^1, \ldots, h_t^n)$. Then, each decoder employs a separate MLP head to produce its respective output. For the centralized critic, we use a separate Transformer encoder to encode the joint deterministic state $h_t^{1:n}$, then map the embedding to a value estimate for each agent.

We also adopt several tricks from DreamerV3, such as the symexp twohot loss for the reward decoder and critic, and free bits for the dynamics loss and representation loss. The reward decoder and critic use the symexp twohot loss as DreamerV3 (Hafner et al., 2025). To be specific, the outputs of reward decoder and critic can be represented as the weighted average of exponentially spaced bins, e.g., $\hat{r}_t^i = \text{Softmax}(\text{MLP}(h_t^i, z_t^i))^\top B$ where $B = \text{symexp}(-20, \ldots, +20)$ and $\text{symexp}(x) = \text{sign}(x)(\exp(|x|) - 1)$. The reward decoder and critic are trained to match the two-hot target using cross-entropy loss.

# D. Hyperparameters

## D.1. Hyperparameters for DMAWM

The empirical results of our DMAWM implementation is based on the hyperparameters in Table 2.

*Table 2.* Hyperparameters for the DMAWM algorithm.

| Hyperparameter | Value |
| --- | --- |
| **Reinforcement Learning** | |
| Optimizer | Adam |
| Entropy coefficient $\beta_{\mathrm{ent}}$ | 0.01 |
| PPO epochs | 5 |
| Clip param $\epsilon$ | 0.2 |
| Actor learning rate | $3 \times 10^{-5}$ |
| Critic learning rate | $3 \times 10^{-5}$ |
| Off-policy critic loss coefficient $\beta_{\mathrm{off}}$ | 1.0 |
| Truncation threshold $C$ | 4 |
| Discount factor $\gamma$ | 0.99 |
| GAE lambda $\lambda$ | 0.95 |
| **World Model** | |
| Max grad norm | 100 |
| Model learning rate | $1 \times 10^{-4}$ |
| Model batch size | 16 |
| Sequence length | 64 |
| Rollout horizon | 4 |
| Buffer size | $2.5 \times 10^5$ |
| KL balancing entropy weight | 0.1 |
| KL balancing cross entropy weight | 1.0 |
| Discrete latent dimensions | 32 |
| Discrete latent classes | 32 |
| Transformer layers | 3 |
| Transformer heads | 8 |
| Decoder hidden size | 1024 |
| Decoder layers | 2 |

### D.2. Hyperparameters for baselines

The experimental results on MAPPO (Yu et al., 2022) is based on the official implementation[2] with the following hyperparameters in Table 3.

*Table 3.* Hyperparameters for the MAPPO algorithm.

| Hyperparameter | Value |
| --- | --- |
| Use RNN | True |
| Optimizer | Adam |
| Episode length | 400 |
| Entropy coefficient | 0.01 |
| Discount factor | 0.99 |
| GAE lambda | 0.95 |
| Critic learning rate | $5 \times 10^{-4}$ |
| Actor learning rate | $5 \times 10^{-4}$ |
| PPO epochs | 5 |
| Clip param | 0.2 |
| Parallel workers | 8 |
| Max grad norm | 10 |

[2]https://github.com/marlbenchmark/on-policy

The experimental results on MAT (Yu et al., 2022) is based on the implementation[3] with the following hyperparameters in Table 4.

*Table 4.* Hyperparameters for the MAT algorithm.

| Hyperparameter | Value |
|---|---|
| Use RNN | True |
| Optimizer | Adam |
| Episode length | 400 |
| Entropy coefficient | 0.01 |
| Discount factor | 0.99 |
| GAE lambda | 0.95 |
| Critic learning rate | $5 \times 10^{-4}$ |
| Actor learning rate | $5 \times 10^{-4}$ |
| PPO epochs | 5 |
| Clip param | 0.2 |
| Parallel workers | 8 |
| Max grad norm | 10 |
| Transformer layers | 1 |
| Transformer heads | 1 |

The experimental results on QMIX (Rashid et al., 2018) is based on the optimized implementation of PyMARL2[4] with the following hyperparameters in Table 5.

*Table 5.* Hyperparameters for the QMIX algorithm.

| Hyperparameter | Value |
|---|---|
| Use RNN | True |
| Optimizer | Adam |
| Learning rate | 0.001 |
| Discount factor | 0.99 |
| Target update interval (episodes) | 200 |
| Max grad norm | 10 |
| Batch size | 128 |
| Buffer size (episodes) | 5000 |
| Epsilon | $1.00 \rightarrow 0.05$ |
| TD lambda | 0.6 |

The experimental results on MAMBA (Egorov & Shpilman, 2022) is based on the official implementation[5] with the following hyperparameters in Table 6.

---

[3] https://github.com/marlbenchmark/on-policy
[4] https://github.com/hijkzzz/pymarl2
[5] https://github.com/jbr-ai-labs/mamba

*Table 6.* Hyperparameters for the MAMBA algorithm.

| Hyperparameter | Value |
|---|---|
| **Reinforcement Learning** | |
| Optimizer | Adam |
| Entropy coefficient | 0.001 |
| Number of updates | 4 |
| PPO epochs | 5 |
| Clip param | 0.2 |
| Actor learning rate | $5 \times 10^{-4}$ |
| Critic learning rate | $5 \times 10^{-4}$ |
| Discount factor | 0.99 |
| GAE lambda | 0.95 |
| **World Model** | |
| Model learning rate | $2 \times 10^{-4}$ |
| Model epochs | 60 |
| Model batch size | 40 |
| Sequence length | 20 |
| Rollout horizon | 15 |
| Buffer size | $2.5 \times 10^{5}$ |
| KL balancing entropy weight | 0.2 |
| KL balancing cross entropy weight | 0.8 |
| Max grad norm | 100 |
| Trajectories between updates | 1 |

The experimental results on MBVD (Xu et al., 2022) is based on the official implementation submitted to OpenReview[6] with the following hyperparameters in Table 7.

*Table 7.* Hyperparameters for the MBVD algorithm.

| Hyperparameter | Value |
|---|---|
| **Reinforcement Learning** | |
| Optimizer | RMSProp |
| Learning rate | $5 \times 10^{-4}$ |
| Discount factor | 0.99 |
| Target update interval (episodes) | 200 |
| Max grad norm | 10 |
| Batch size | 32 |
| Buffer size (episodes) | 5000 |
| Epsilon | $1.00 \rightarrow 0.05$ |
| **World Model** | |
| Rollout horizon | 3 |
| KL balancing entropy weight | 0.3 |
| KL balancing cross entropy weight | 0.7 |
| Trajectories between updates | 1 |

---

[6]https://openreview.net/forum?id=flBYpZkW6ST

# E. Environment descriptions

### E.1. SMAC

StarCraft Multi-Agent Challenge (SMAC) (Samvelyan et al., 2019) is a popular benchmark for MARL research based on the real-time strategy game StarCraft II. It offers a collection of micro battle scenarios in StarCraft II, where a team of ally units must collaborate to defeat the opposing team controlled by rule-based bots. In these scenarios, each agent is responsible for controlling one ally unit and has access to information such as the distance, relative location, health, shield, and type of both ally and enemy units within their field of vision. For our purposes, we consider each unit as an entity, with ally units categorized as agent entities and enemies as non-agent entities.

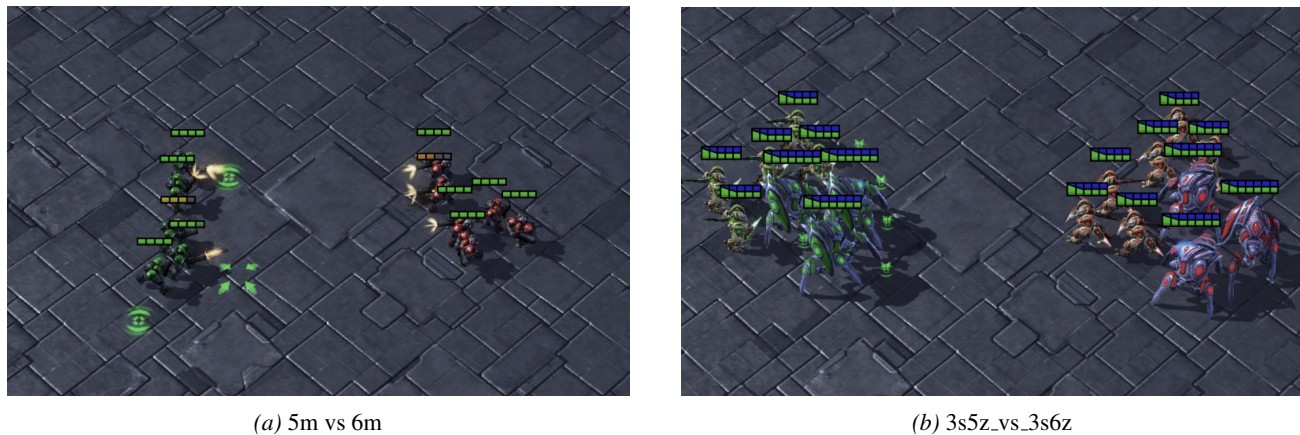

*(a)* 5m vs 6m        *(b)* 3s5z_vs_3s6z

*Figure 8.* SMAC

### E.2. SMACv2

SMACv2 (Ellis et al., 2023) extends SMAC by introducing increased complexity and randomness. It randomizes the starting positions and unit types of agents with varying sight and attack ranges, presenting MARL algorithms with greater levels of stochasticity and diversity. Similar to the approach taken in SMAC, we consider each unit as an individual entity, ally units as agent entities, and enemies as non-agent entities.

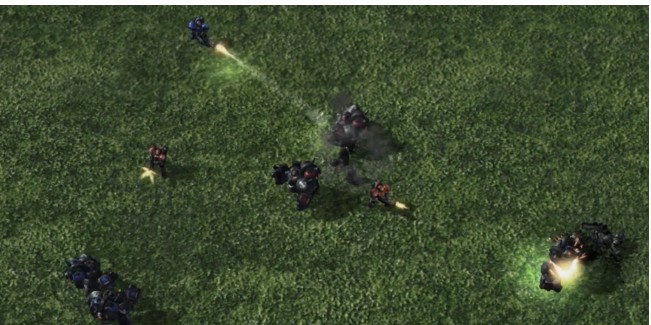

*Figure 9.* SMACv2

### E.3. Melting Pot

Melting Pot (Leibo et al., 2021) provides a suite of multi-agent tasks and an evaluation protocol for assessing the social intelligence of agents. These tasks are vision-based, where the observations are ego-centric 2D visual observations of the environment.

Coop Mining (Figure 10a), a cooperative scenario where agents coordinate to collect resources, is an instance of the cooperative task in Melting Pot. The environment features two resource types: iron ore and gold ore. Iron ore can be

gathered by a single agent, but gold ore necessitates the use of beams by two agents within a time window of 3 timesteps. Collecting iron ore yields a reward of 1 for the agent, while successfully gathering gold ore grants a reward of 8 to each participating agent. Each episode lasts 1000 timesteps.

Common Harvest (Figure 10b), in which agents consume renewable common resources, is a tragedy-of-the-commons scenario. Apples are initially scattered throughout the environment, and consuming one yields a reward of 1. At each timestep, apples respawn with a probability that is positively correlated with the number of apples within a neighborhood of radius 2. Consequently, an isolated patch (one with no other apples within distance 2) can be permanently depleted if all apples in that patch are consumed. Agents must therefore exercise restraint when consuming apples within a patch. Each episode lasts 1000 timesteps.

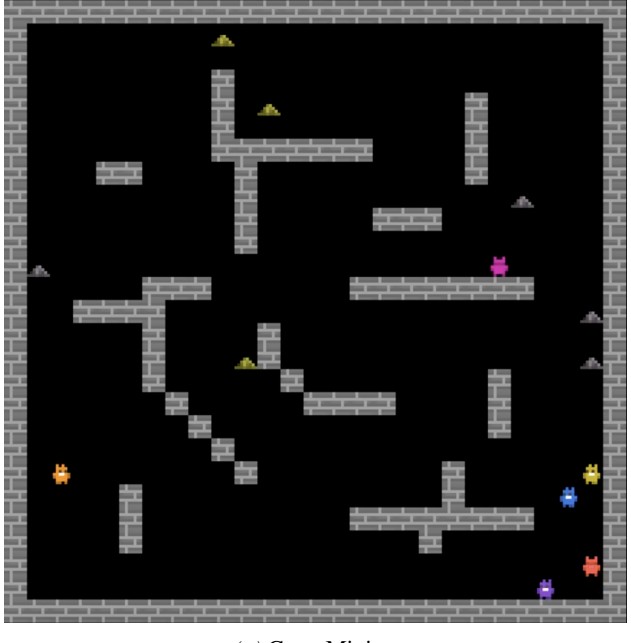

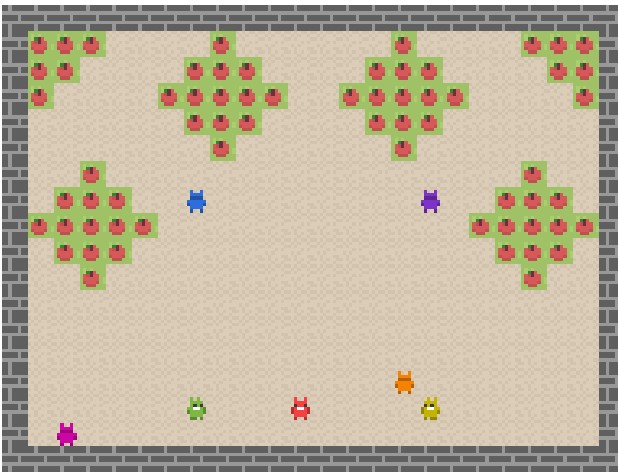

(a) Coop Mining                                   (b) Common Harvest

*Figure 10.* Melting Pot

To save GPU memory, we resize the observation from $88 \times 88$ to $44 \times 44$. We show an example of the original and resized observations in Figure 11 below.

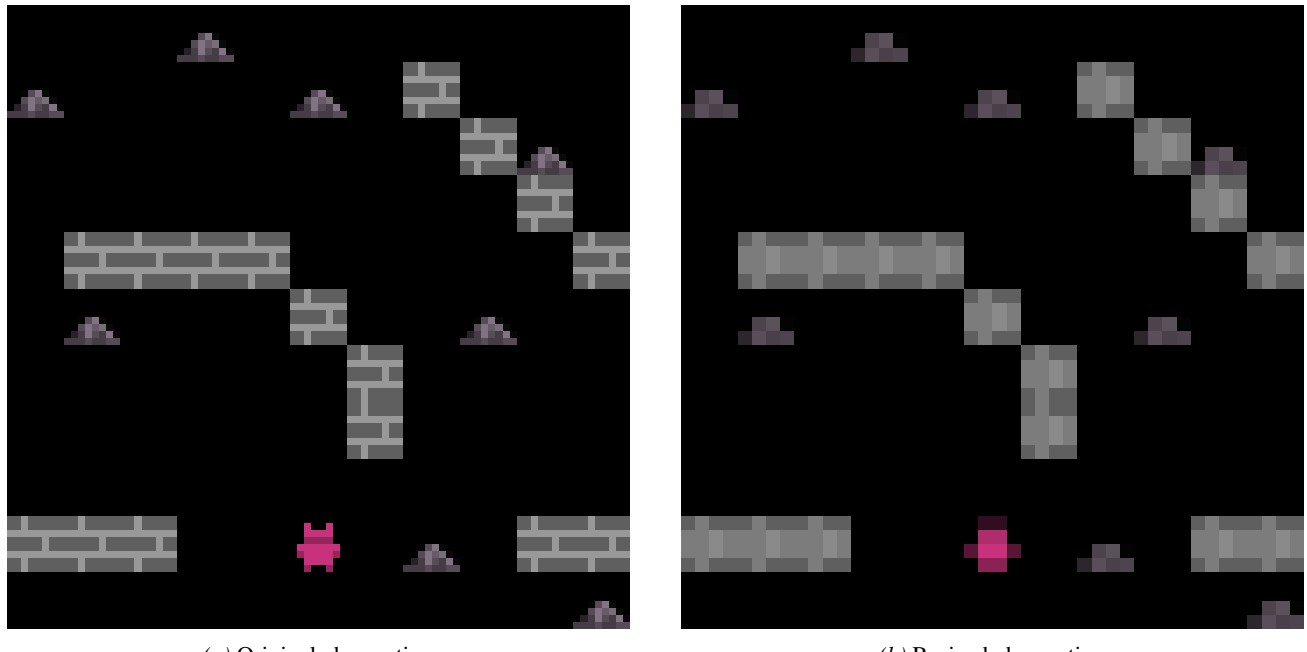

*(a)* Original observation                    *(b)* Resized observation

*Figure 11.* Observation Resizing

# F. More wall-clock-time results

To evaluate the runtime efficiency of DMAWM, we measure the wall-clock-time of DMAWM on the SMAC and SMACv2 benchmark using a single NVIDIA RTX 3090 GPU. The results are shown in Figure 12.

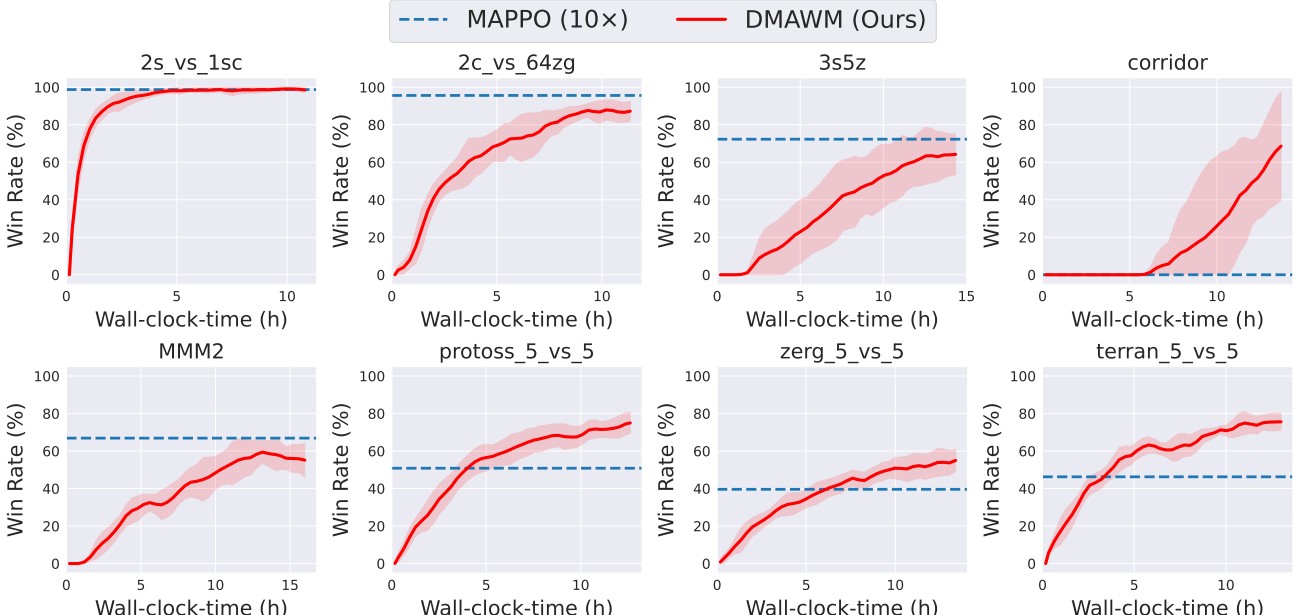

*Figure 12.* Training wall-clock-time of DMAWM on the SMAC and SMACv2 benchmarks. Results are averaged over 3 independent runs. DMAWM is trained for 400K environment steps, while MAPPO is trained for 4M environment steps.

## G. Comparison with MARIE

To compare with MARIE (Zhang et al., 2025), we evaluate DMAWM on the benchmark used in the original MARIE paper, following the same experimental protocol (evaluation metrics, number of runs, and training steps).

*Table 8.* Comparison on MARIE's benchmark. We report the win rate (%) with standard deviation on SMAC maps used in Zhang et al. (2025).

| Maps | Difficulty | Steps | DMAWM (Ours) | MARIE |
|---|---|---|---|---|
| 1c3s5z | | | 75.0(13.3) | **85.0**(9.4) |
| 2m_vs_1z | | | **99.9**(0.4) | 95.5(7.9) |
| 2s_vs_1sc | | | 95.7(4.3) | **96.9**(7.1) |
| 2s3z | | | **87.8**(4.2) | 80.5(9.3) |
| 3m | *Easy* | 100K | **99.5**(1.2) | **99.5**(0.4) |
| 3s_vs_3z | | | 98.8(1.8) | **98.9**(1.5) |
| 3s_vs_4z | | | **94.8**(3.5) | 73.0(6.2) |
| 8m | | | **95.5**(3.6) | 88.0(3.9) |
| MMM | | | **94.3**(2.7) | 87.6(3.0) |
| so_many_baneling | | | **97.7**(2.2) | 94.8(5.9) |
| 3s_vs_5z | *Hard* | 200K | **89.1**(5.2) | 78.4(11.2) |
| 2c_vs_64zg | | | **76.6**(9.7) | 25.9(14.3) |
| corridor | *SuperHard* | 400K | **71.5**(25.9) | 71.0(13.8) |

## H. Scalability to 20 Agents

We further evaluate DMAWM on SMACv2 scenarios with 20 agents to study scalability beyond the 5-agent settings in the main experiments. As shown in Table 9, DMAWM remains competitive as the number of agents increases, substantially outperforming MAMBA and MAPPO on all three maps. The per-agent modules, including the agent module and actor, scale linearly with the number of agents. The environment module and centralized critic apply self-attention over the joint agent states and therefore have quadratic complexity in the number of agents, but this overhead remains manageable in the tested 20-agent scenarios.

*Table 9.* Scalability results on 20-agent SMACv2 maps. We report the win rate (%) with standard deviation.

| Map | Steps | DMAWM (Ours) | MAMBA | MAPPO |
|---|---|---|---|---|
| protoss_20_vs_20 | 400K | **56.6**(5.1) | 4.3(2.4) | 1.0(1.2) |
| terran_20_vs_20 | 400K | **52.8**(4.3) | 4.5(2.4) | 0.0(0.0) |
| zerg_20_vs_20 | 400K | **12.5**(5.7) | 1.9(0.8) | 0.0(0.0) |

## I. Extended training results

While the main experiments compare all algorithms at 400K environment steps for fair comparison with baselines, we additionally train DMAWM for 1M steps to investigate its asymptotic performance and convergence behavior. The results are shown in Figure 13.

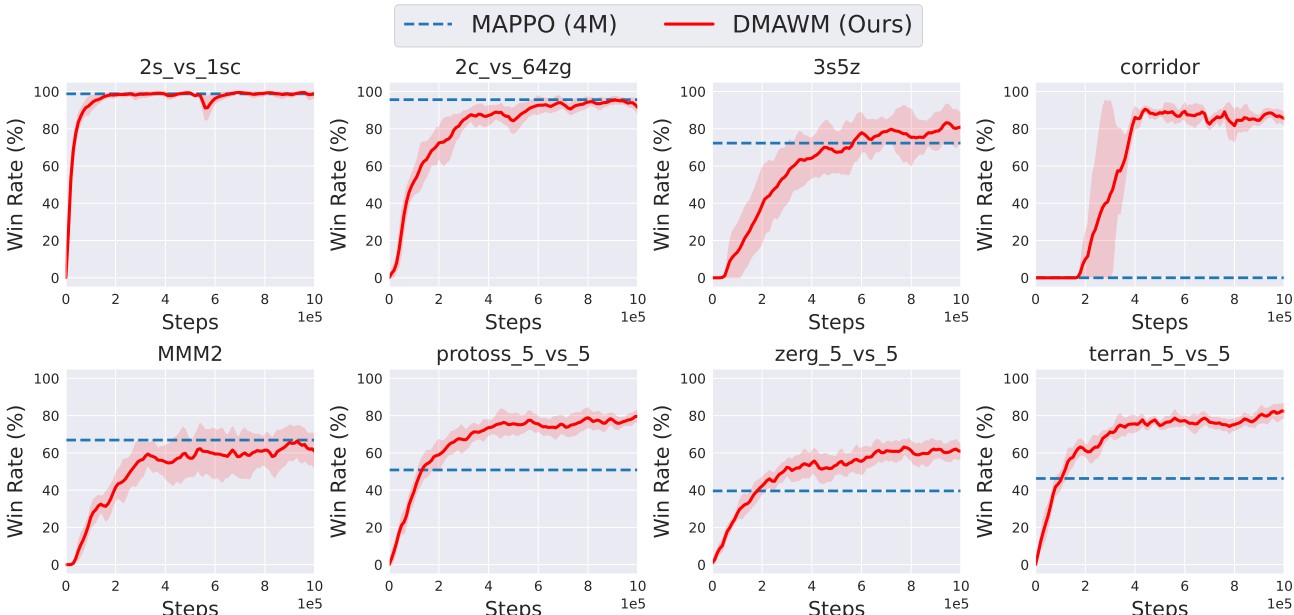

*Figure 13.* Training curves of DMAWM extended to 1M environment steps. Results are averaged over 3 independent runs, with shaded areas representing the standard deviation. Dashed lines show performance of MAPPO training for 4M environment steps.

## J. Model error analysis

In Table 10, we provide an evaluation of the dynamics model by measuring the cumulative observation reconstruction error over varying imaginary rollout lengths on the Coop Mining map. We use the mean squared error (MSE) to measure the reconstruction error. The results show that the error remains low for the first 16 steps. The increase of error at longer horizons is expected due to the environment's stochasticity (randomized ore positions), which aligns with the visualization in Figure 7. Since we choose the imagination rollout length to be 4 frames, we are in the regime where the model error is relatively small.

*Table 10.* Cumulative observation reconstruction error (MSE) over varying imaginary rollout lengths on the Coop Mining map.

| Rollout steps | 4 | 8 | 12 | 16 | 20 | 24 | 28 | 32 | 36 | 40 |
|---|---|---|---|---|---|---|---|---|---|---|
| Cumulative reconstruction error | 0.0215 | 0.0392 | 0.0752 | 0.1160 | 0.1820 | 0.2599 | 0.3585 | 0.4560 | 0.5784 | 0.6943 |
| Difference | 0.0215 | 0.0177 | 0.0360 | 0.0408 | 0.0660 | 0.0779 | 0.0986 | 0.0975 | 0.1224 | 0.1159 |

## K. Limitations

**Short imagination horizon.** As shown in Figure 5(b), DMAWM's performance does not improve beyond short imagination horizons (4 steps). We attribute this plateau to compounding model errors, which are amplified in multi-agent settings where novel joint actions are more likely to cause imagined trajectories to diverge from real ones. This limitation is consistent with observations in single-agent model-based RL: the original Dreamer reported marginal improvements beyond short horizons, and TD-MPC deliberately uses only 3-step rollouts. While this remains an open challenge in the field, DMAWM achieves high sample efficiency even with these short horizons.

**Parameter sharing.** DMAWM shares parameters across all agent modules, which limits applicability to heterogeneous teams where agents have different observation spaces, action spaces, or capabilities. For moderate heterogeneity (e.g., different unit types in SMAC), this issue can be mitigated by appending one-hot type encodings to observations and padding action spaces to a common length. For settings with fundamentally different agent capabilities, DMAWM can instantiate separate agent modules for each agent type while keeping the shared environment module and disentanglement objective unchanged.

