# OpenReview forum: "Learning Disentangled Multi-Agent World Model for Decentralized Control"
_ICML.cc/2026/Conference — ICML 2026 regular_

### Official Review · Reviewer_bbdZ · 2026-03-13

**Soundness:** 3
**Presentation:** 4
**Significance:** 4
**Originality:** 3
**Overall Recommendation:** 5
**Confidence:** 4

**Summary:**

This paper proposes an adaptation of the Dreamer algorithm to the collaborative multiagent setting.
Adapting this algorithm is motivated by the fact that it allows for latent imagination and policy optimization, which is efficient.
Previous attempts have either relied on observation reconstruction (losing this efficiency), or have relaxed assumptions by allowing communication between the agents, preventing fully decentralized execution.
This paper proposes a principled method for having a joint latent imagination (that is claimed to be able to correctly model agent interactions), while still allowing fully decentralized execution.
The empirical results show very good performance compared to related methods.

**Compliance With Llm Reviewing Policy:**

Affirmed.

**Final Justification:**

The authors answered all my questions and I note that the authors have also addressed my minor concerns. My final recommendation is to accept the paper.

**Key Questions For Authors:**

- From figure 5b, you claim that DMAWM is robust to the choice of the imagination horizon. However, the fact that the performance is constant when the horizon grows could also highlight the fact that the methods does not scale to longer effective horizons since it does not succeed in learning better policies. What are you thoughts on that?

**Strengths And Weaknesses:**

Correctly adapting Dreamer to the multiagent setting is an important contribution to the community.
I believe that this paper has done it right, and I recommend acceptance.
Moreover, the paper is very well structured and pleasant to read.

- *Soundness*. I have two remarks regarding the modeling choices.
    - Loss of tightness of the ELBO. While it is required for decentralized execution, I think that it would be a good idea to discuss the fact that the choice of posterior (the product of marginal of the agent encoders) is not as expressive as the posterior that a centralized dreamer would have used. The direct consequence of restricting the posterior $q_\psi(z_t | h_t, o_t)$ to $\Pi_{i=1}^n q_\psi(z_t^i | h_t^i, o_t^i)$ is that the ELBO may not necessarily become a tight lower bound on the log likelihood. I agree that during optimization, the ELBO will rarely be tight in practice, but it is worth noting that here, even if you had an oracle to directly find the best posterior in this hypothesis class (of factorized posteriors), the ELBO evaluated with your factorized posterior would not correspond to the likelihood.
    - Limited expressiveness of the interaction predictor (i.e., the prior). By selecting a transformer that takes as input all $h_t^i$ and outputs the distribution parameters of all $z_t^i$ (which I suppose are the logits because you probably use discrete latent variables), you are implicitly making a conditional independence assumption. Indeed, all $z_t^i$ will be conditionally independent of each others given all $h_t^i$. This is not a strong modeling assumption since it is conditional on all $h_t^i$ jointly. That said, you might have wanted to use an autoregressive Transformer instead, where the distribution of each $z_t^i$ is conditioned on $h_t^{1:n}$ and $z_t^{\lt i}$. This is an overhead that would make the sampling procedure iterative (slower), and I am not sure it would result in a better performance. But it may be interesting to discuss it.
- *Presentation*. I found the presentation to be very good.
    - One suggestion would be to better link the fact that the "interaction predictor" corresponds to the prior in the standard dreamer algorithm in section 4.1.
    - I would suggest to better explain the meaning of $m_t^{1:n}$ and how it is used in the action sampling procedure and/or policy optimization procedure.
- *Significance*. Having such a clean and principled dreamer-like algorithm for collaborative multiagent RL is definitely useful for the community.
- *Originality*. Most of the technical novelty is to adapt dreamer to the decentralized setting. One additional novelty is the off-policy loss.

---

> ### Author Rebuttal · Authors · 2026-03-31
>
> We thank the reviewer for the insightful comments on our method. We clarify each point below.
>
> > W1: Loss of tightness of the ELBO
>
> We appreciate the reviewer for pointing out this nuance.
> We respectfully clarify that this factorized posterior is a direct consequence of the architectural design of the agent module, rather than a deliberate choice made for disentanglement.
> Because the individual stochastic latent state depends strictly on its local deterministic state and observation $z\_t^i\sim q\_\psi(z\_t^i \mid h\_t^i, o\_t^i)$, the joint posterior naturally factorizes as $q\_\psi(z^{1:n}\_t \mid h\_t^{1:n}, o\_t^{1:n}) = \prod\_{i=1}^n q\_\psi(z\_t^i \mid h\_t^i, o\_t^i)$.
> As a consequence, the disentanglement objective itself would not make the ELBO less tight.
> We will revise the manuscript to explicitly include this equation and clarification.
>
> > W2: Limited expressiveness of the interaction predictor (i.e., the prior)
>
> You are correct that our current interaction predictor makes an implicit conditional independence assumption, as we assume the joint deterministic states $h\_t^{1:n}$ encode enough context for capturing the necessary dependencies among agents.
>
> Actually, we have experimented replacing the interaction predictor with an autoregressive transformer. However, we found that the primary bottleneck is not the computational overhead but the significant training instability. We hypothesize that this instability stems from two factors: (1) The autoregressive formulation expands the input space and breaks the permutation-invariance, which significantly increases the learning complexity. (2) The sequential sampling process introduces compounded variance on the latent states, making the imaginary trajectories overly stochastic.
>
> Therefore, we choose our current design to trade expressiveness for training stability due to these practical considerations.
> Following your suggestion, we would like to make this consideration explicit and add a discussion of this in the appendix.
>
> > W3: Presentation issues
>
> We sincerely thank the reviewer for these helpful suggestions. We will make sure to update Section 4.1 to explicitly relate the output of the interaction predictor to the standard Dreamer prior, and elaborate on the usage of the available action mask in both action sampling and policy optimization procedures.
>
> > Q1: From figure 5b, you claim that DMAWM is robust to the choice of the imagination horizon. However, the fact that the performance is constant when the horizon grows could also highlight the fact that the methods does not scale to longer effective horizons since it does not succeed in learning better policies. What are you thoughts on that?
>
> We thank the reviewer for pointing this out, and we agree that it points to a limitation of the method. We will revise the discussion of Figure 5b to explicitly acknowledge this.
>
> This plateau in performance is likely due to compounding model errors over long horizons, which is further amplified in multi-agent environments.
> In particular, interaction among multiple agents makes it more likely that a novel joint action causes imagined trajectories to diverge from real ones.
>
> This limitation is widely acknowledged in the model-based RL literature. For instance, the original dreamer paper [1] reported marginal improvements for longer imagination horizons, and methods like TD-MPC [2, 3] deliberately utilize very short horizons (only 3 steps).
> While this remains an open challenge in the field, the good news is that DMAWM could achieve high sample efficiency even with very short imagination horizons. From a practical standpoint, this allows our method to strike a balance between sample efficiency and wall-clock-time.
>
> References:
> - [1] Dream to Control: Learning Behaviors by Latent Imagination. ICLR 2020.
> - [2] Temporal Difference Learning for Model Predictive Control. ICML 2022.
> - [3] TD-MPC2: Scalable, Robust World Models for Continuous Control. ICLR 2024.

---

> > ### Author Rebuttal · Reviewer_bbdZ · 2026-03-31
> >
> > Thank you for your answer, and in particular for discussing the results of the autoregressive version. I would like to reiterate my point about the ELBO, sorry if that was not clear the first time.
> >
> > Let us consider a slightly simplified setting, without recurrence. We have, for any latent variable model $q_\phi(o, z) = q_\phi(z) q_\phi(o|z)$ and any encoder $q_\psi$,
> > $$
> >     \operatorname{ELBO}(o) = E_{z \sim q_\psi(z|o)}[ \log q_\phi(o|z) ] - KL_z(q\_\psi(z|o) \parallel q_\phi(z)) \leq q_\phi(o),
> > $$
> > where the likelihood $q_\phi(o)$ is obtained by marginalization of the latent variable model $q_\phi(o, z)$. And, we also have, for any encoder $q_\psi$,
> > $$
> >     \operatorname{ELBO}(o) = \log q_\phi(o) - KL_z(q_\psi(z|o) \parallel q_\phi(z|o))
> > $$
> > where $q_\phi(z|o)$ is the *true* posterior of the latent variable model $q_\phi(o, z)$. As can be seen from this expression, the tightness of the ELBO is measured by the KL between the *true* (untractable) posterior $q_\phi$ and the *approximate* (learned) variational posterior $q_\psi$. The hypothesis class of the variational posterior should thus contain the *true* posterior in order to make the ELBO tight, which is not necessarily the case in your paper.
> >
> > To answer your rebuttal, I fully agree that posing $z^i \sim q_\psi(z^i | o^i)$ is equivalent to posing $q_\psi(z | o) = \prod_{i=1}^n q_\psi(z^i | o^i)$. Thus, I agree that your posterior is factorizable, by definition/assumption. This is not the problem. The problem is that the *true* posterior $q_\phi(z | o)$ is not necessarily factorizable in a product of marginals.
> >
> > Let us take a simple example where the joint prior $q_\phi(z)$ selects $z = [z^1, z^2] = [0, 0]$ with a probability 0.5, and $z = [z^1, z^2] = [1, 1]$ with probability 0.5. Let us assume that $q_\phi(o|z)$ outputs $o = z = [z^1, z^2]$ with probability 0.95, and $o = [z^1, 1 - z^2]$ with probability 0.05. The *true* posterior $q_\phi(z|o=[0, 1])$ is a Dirac at $z = [0, 0]$ while the best factorized variational posterior $q_\psi(z|o=[0, 1]) = q_\psi(z^1|o^1=0) q_\psi(z^2|o^2=1)$ gives a probability of 0.5 for $z = [0, 0]$ and a probability of 0.5 for $z = [0, 1]$, while the latter sample is not even in the support of the *true* posterior distribution.

---

> > > ### Author Response · Authors · 2026-04-01
> > >
> > > We sincerely thank the reviewer for the detailed clarification and example. We now fully understand the concern, and apologize that our initial response did not address it precisely.
> > >
> > > The reviewer's point is that the tightness of ELBO is governed by $\text{KL}(q\_\psi(z|o) \| q\_\phi(\hat{z}|o))$, where $q\_\phi(\hat{z}|o)$ is the true posterior of the generative model.
> > > In our case, this KL divergence is instantiated as $\text{KL}(q\_\psi(z\_t^{1:n}| h\_t^{1:n}, o\_t^{1:n}) \| q\_\phi(\hat{z}\_t^{1:n}| h\_t^{1:n}, o\_t^{1:n}))$. We acknowledge that in the general case, restricting the variational family to factorized distributions would make the ELBO less tight, because there are dependencies that a joint $q\_\psi(z\_t^{1:n}| h\_t^{1:n}, o\_t^{1:n})$ could capture but a factorized $\prod\_{i=1}^n q\_\psi(\hat{z}\_t^i \mid h\_t^i, o\_t^i)$ cannot.
> > >
> > > However, we would like to highlight that we optimize the KL objective (Equation 4) in both directions.
> > > The dynamics loss $\mathcal{L}\_{\rm dyn}$ and the representation loss $\mathcal{L}\_{\rm rep}$ jointly trains the prior $p\_\phi(\hat{z}^{1:n}\_t \mid h^{1:n}\_t)$ and the factorized posterior $\prod\_{i=1}^n q\_\psi(z^i\_t \mid h^i\_t, o^i\_t)$ to approximate each other.
> > > As the prior and the posterior are more aligned, the ELBO objective could be effectively optimized, which also pushes the true posterior to be more aligned with the factorized posterior.
> > >
> > > Upon closer examination of our architecture, we find that the true posterior of our generative model is actually factorizable across agents. As the reviewer identified that our prior is conditionally independent, i.e., $p\_\phi(\hat{z}\_t^{1:n}\mid h\_t^{1:n}) = \prod\_{i=1}^n p\_\phi(\hat{z}\_t^i\mid h\_t^{1:n})$, the corresponding true posterior factorizes as:
> > > $$
> > > p\_\phi(\hat{z}\_t^{1:n}\mid h\_t^{1:n}, o\_t^{1:n}) \propto p\_\phi(o\_t^{1:n}\mid h\_t^{1:n}, \hat{z}\_t^{1:n}) p\_\phi(\hat{z}\_t^{1:n}\mid h\_t^{1:n}) = \prod\_{i=1}^n p\_\phi(o\_t^i\mid h\_t^i, \hat{z}\_t^i) p\_\phi(\hat{z}\_t^i\mid h\_t^{1:n}) ,
> > > $$
> > > where the observation decoder is local to each agent.
> > > We assume this factorizability could make the alignment between the true and factorized posteriors easier for our architecture.
> > >
> > > In summary, while a factorized posterior would lose tightness in the general case, the bi-directional KL objective could mitigate this issue.
> > > We are grateful to the reviewer for deepening our understanding of this aspect of the method, and hope this will address your concern.

---

### Official Review · Reviewer_AeQw · 2026-03-13

**Soundness:** 3
**Presentation:** 3
**Significance:** 4
**Originality:** 2
**Overall Recommendation:** 5
**Confidence:** 4

**Summary:**

This paper proposes DMAWM, a model-based MARL framework that learns decentralized policies in latent space. It separates the world model into independent agent modules and a shared environment module with a transformer-based interaction predictor. KL divergence alignment between joint prior and factorized posterior enforces disentanglement, in a way that imagined rollouts respect decentralized execution. Experiments on SMAC, SMACv2, and Melting Pot show strong improvements over baselines, and a further experiment confirmed that the latent states avoid information leakage.

**Compliance With Llm Reviewing Policy:**

Affirmed.

**Final Justification:**

I thank the authors for the thorough rebuttal. The new 20-agent experiments on SMACv2 directly address my scalability concern and the results are convincing. The MATWM comparison strengthens the empirical contribution, and the Latent MAPPO ablation cleanly separates the gains from imagination versus architecture, which was my main curiosity. The short horizon discussion is reasonable, though I still encourage the authors to include an explicit limitations section covering this point in the final version. Overall, the rebuttal has resolved my main concerns and I maintain my acceptance recommendation.

**Key Questions For Authors:**

1. If I am right, the largest environment you used is the MMM2, with 10 agents. Do youu have ideas of how DMAWM scale to 15+ agents? It is not necessary, but can you provide cost analysis or experiments? Or at least a mention about it, some insight could help future research and use of this paper.
2. Could the disentanglement constraint be too restrictive for tightly coupled coordination tasks, forcing the model to ignore important correlations?
3. How much benefit comes from imagination vs architectural design? An ablation with 0-step imagination using same architecture and model-free training would clarify this. On fig 5b you have started on 4.

**Limitations:**

I would recommend to include an explicitly limitations section, and if possible write about the use of a short world model horizon in it.

**Strengths And Weaknesses:**

Strengths:
1. Well-identified problem: latent world models create spurious inter-agent correlations incompatible with decentralized execution. The proposed solution, aligning a joint prior with factorized posteriors, is principled via an ELBO derivation.
2. Strong empirical results across three benchmarks, with particularly large margins on Melting Pot and SMACv2. Wall-clock comparisons and MARIE comparison add credibility.
3. The experiment on section 5.4 convinced me that DMAWM avoids information leakage: 42.3% vs MAMBA's 85.0% for unobservable teammates.
4. Thorough ablations covering all key components, imagination horizon sensitivity, extended training, and model error analysis.

Weaknesses:
1. From my point of view my biggest complain is about the very short imagination horizon adopted (4 steps) in comparison with MAMBA and MARIE. Seems like performance does not improve with longer horizons on Fig 5b. This raises questions about how much benefit actually comes from imagination vs. architectural choices.
2. The comparisons are fair, but a little "time-biased", most of the methods used for comparison are already a little old. The comparison against MARIE was a great point. However I missed more robust and recent baselines appearing on the comparison tables, MATWM for example.
3. Parameter sharing limits generality. All agent modules share parameters, restricting applicability to heterogeneous settings. This is not clearly discussed.

---

> ### Author Rebuttal · Authors · 2026-03-31
>
> We sincerely thank the reviewer for the positive assessment and the constructive suggestions. We provide new experiments on scalability, additional baselines, and ablations below.
>
> > W1: Performance does not improve with longer horizons.
>
> We agree this points to a limitation and will revise the discussion of Figure 5b accordingly. We attribute this plateau to compounding model errors, which is amplified in multi-agent settings as novel joint actions are more likely to cause imagined trajectories to diverge from real ones. Reviewer bbdZ also raises this issue, and we respectfully point the reviewer there (Q1) for a more thorough analysis due to space constraints.
>
> > W2: Missing comparison with more recent work, e.g., MATWM.
>
> Since the MATWM GitHub repository is currently inaccessible, we evaluate DMAWM directly on the subset of SMAC maps used in the MATWM benchmark with the same hyperparameters as our main experiments. DMAWM outperforms MATWM on 11 out of 12 maps, and we show a subset of the results due to the character limitation. We will add a discussion of MATWM to the related works Section.
>
> | Map | Steps | DMAWM (Ours) | MATWM |
> | - | - | - | - |
> | 2s3z | 50K | 76.9(13.7) | **80.0(9.0)** |
> | 8m | 50K | **93.8(2.4)** | 67.0(24.9) |
> | MMM | 50K | **85.0(2.7)** | 7.0(4.7) |
> | 3s\_vs\_5z | 200K | **89.1(5.2)** | 64.0(26.5) |
> | 2c\_vs\_64zg | 200K | **76.6(9.7)** | 7.0(7.5) |
> | 5m\_vs\_6m | 200K | **58.9(9.5)** | 46.0(21.8) |
>
> > W3: Parameter sharing limits applicability in heterogeneous settings.
>
> We chose parameter sharing primarily for sample efficiency, but agree it limits flexibility in heterogeneous settings.
> To address the heterogeneity within the current setup, the most straightforward approach is appending one-hot type encodings to the observations and padding action-space to the same length. For settings with fundamentally different agent capabilities, DMAWM can naturally instantiate separate agent modules for different agent types while keeping the shared environment module and the disentanglement objective unchanged.
>
> > Q1: Scalability to 15+ agents?
>
> To directly address your question, we conducted additional experiments on SMACv2 using 20 agents.
> As shown in the table below, although the win rate on `zerg_20_vs_20` is lower than the other maps, the overall performance of DMAWM remains highly competitive as the number of agents scales.
>
> | Map | Step | DMAWM (Ours) | MAMBA | MAPPO |
> | - | - | - | - | - |
> | protoss\_20\_vs\_20 | 400K | **56.6(5.1)** | 4.3(2.4) | 1.0(1.2) |
> | terran\_20\_vs\_20 | 400K | **52.8(4.3)** | 4.5(2.4) | 0.0(0.0) |
> | zerg\_20\_vs\_20 | 400K | **12.5(5.7)** | 1.9(0.8) | 0.0(0.0) |
>
> For the computational cost, the per-agent modules, such as the agent module and the actor, scale linearly with the number of agents. The environment module and the centralized critic incur quadratic complexity with respect to the number of agents, as they apply self-attention to the joint agent states. However, we found that this overhead remains manageable for environments with up to 20 agents.
>
> Scaling beyond 20 agents presents non-trivial challenges to model-based methods. One promising direction is to exploit the agent topology (e.g., networked MDP) to simplify the inter-agent interactions that should be captured by the world model.
>
> > Q2: Could disentanglement be too restrictive for tightly coupled coordination?
>
> Our disentanglement constraint is designed to prevent unrealistic dependencies that are impossible during decentralized execution, rather than removing all inter-agent dependencies.
> Even though the latent states of agents are conditionally independent given observations, they remained correlated once the observations are not accessible, which is exactly what the joint prior tries to capture.
> As qualitative evidence, the visualization in Appendix I demonstrates that tightly coupled coordination (e.g., relative positions and mining beam actions) is precisely captured by the world model's prior.
>
> > Q3: How much benefit comes from imagination vs. architecture?
>
> To decouple the gains of imagination from our architectural design, we evaluated a variant that shares the same architecture as DMAWM, which we refer to as Latent MAPPO. In this variant, the agent modules and environment module are trained via the world modeling loss, while the actor and critic are purely trained on the on-policy trajectories.
> As shown below, Latent MAPPO only slightly improves over MAPPO, while the full DMAWM significantly outperforms both, demonstrating the importance of latent imagination.
>
> | Map | Step | DMAWM (Ours) | Latent MAPPO | MAPPO |
> | - | - | - | - | - |
> | protoss\_5\_vs\_5 | 400K | **77.0(5.8)** | 24.3(2.4) | 16.9(9.2) |
> | terran\_5\_vs\_5 | 400K | **76.0(3.8)** | 24.5(2.4) | 15.0(4.9) |
> | zerg\_5\_vs\_5 | 400K | **55.3(5.0)** | 11.9(0.8) | 7.9(3.6) |

---

### Official Review · Reviewer_b5QB · 2026-03-13

**Soundness:** 3
**Presentation:** 3
**Significance:** 2
**Originality:** 2
**Overall Recommendation:** 4
**Confidence:** 3

**Summary:**

The paper proposes a Multi-Agent extension to the Dreamer architecture, expanding Dreamer from a POMDP to a Dec-POMDP setting. The key innovation is a transformer-based latent fusion module that captures inter-agent interactions during imagination, trained to eliminate spurious correlations. Experiments conducted on three MARL benchmark sets demonstrate performance improvements over various model-based and model-free baselines. Ablation studies are performed to identify which parts of the architecture are critical for performance and for capturing agent interactions.

**Compliance With Llm Reviewing Policy:**

Affirmed.

**Final Justification:**

The authors have addressed my concerns mostly. As mentioned in my comments, the authors needs to have a more detailed discussion of related works and bring in comparisons with very relevant baselines (several parts of the architecture are directly inherited from previous works these multi agent model based baseline). The main contribution/ finding is the clever/elegant KL loss to avoid spurious correlations and even though a small increment over existing works, seems to impactful for performance. I hence increase my score to a weak accept (4) and increase my subscore for novelty to fair (2).

**Key Questions For Authors:**

1. Could the authors discuss these recent works in the related works section:
[A] https://arxiv.org/pdf/2406.15836 (Decentralized Transformers with Centralised Aggregation are Sample-Efficient Multi-Agent World Models, TMLR 2025),
[B] https://arxiv.org/pdf/2304.06011 (MABL: Bi-Level Latent-Variable World Model for Sample-Efficient Multi-Agent Reinforcement Learning),
[C] https://arxiv.org/pdf/2406.13600 (CoDreamer: Communication-Based Decentralised World Models, RLC Workshop 2024).

2. Could they compare the results with some of these and determine if there are significant performance gaps?

**Limitations:**

Yes, the authors are upfront about a few limitations discussed in the future works section.

**Strengths And Weaknesses:**

Strengths
1. The paper is well written and easy to follow.
2. The paper extends the Dremaer framework in a clear and elegant way to the Dec-POMDP setting.
3. The way in which KL loss was formulated to avoid spurious correlations in the shared latent is elegant.
4. The ablations performed are informative and provide insights into which components are critical for good performance.

Weaknesses
1. I think the novelty is limited, as the main contribution is the latent fusion based on a transformer, which has been previously explored in MARL+World Modelling settings (see the reference list under the questions section).
2. There is insufficient discussion of related works. The paper appears to overlook several recent works that combine Dreamer+MARL or transformer-based information fusion in Multi-Agent World Modelling(see the reference list under the questions section).
3. Similarly, the comparison with these related works in the experiments section should be clarified. If your approach outperforms them, explain why, as the differences between the architectures proposed in these methods and yours seem to have similar motivations. Looking at the experiment section of these references, they seem to closely resemble and in some cases exceed your numbers in SMAC benchmarks [A,B] and [C] have numbers melting pot. All three following dreamer + MARL world modelling storyline.

---

> ### Author Rebuttal · Authors · 2026-03-31
>
> We thank the reviewer for the detailed feedback. We address the concerns about novelty and missing baselines, and provide new experimental results below.
>
> > W1: Limited novelty — transformer-based latent fusion explored before in MARL+World Modelling
>
> We appreciate the concern and would like to clarify the distinction. The novelty of our work lies not in the fusion mechanism, but in the latent disentanglement formulation, which is supported by the architectural design and the disentanglement objective.
>
> Concretely, prior works such as MARIE [A] and CoDreamer [C] focus on designing richer information fusion mechanisms (e.g., Perceiver Transformer, GNN-based communication) to improve joint state modeling. Our work takes the opposite direction: rather than building more expressive fusion, we show that disentangling the latent states is the critical factor for learning environment models suitable for decentralized control. This is supported by the probing experiment in Section 5.4, where DMAWM achieves $42.3\\%$ teammates action prediction accuracy compared to MAMBA's $85.0\\%$, confirming that our objective successfully reduces information leakage between agents.
>
> Moreover, this design enables DMAWM to achieve strong performance with a simpler architecture that strictly adheres to the CTDE paradigm, matching or outperforming transformer-based baselines on 10 out of 13 SMAC maps (Appendix G).
>
> > W2, Q1: Missing related works — [A], [B], [C]
>
> We sincerely thank the reviewer for highlighting these relevant works. We would like to respectfully clarify that all three of these papers are already cited and discussed in Section 2 (related work) of our current submission, but we agree that the distinctions could be stated more clearly. In particular:
> - MARIE [A] is a transformer-based multi-agent world model, where a shared transformer handles decentralized local dynamics, while a Perceiver transformer compresses the joint observation-action sequence into per-agent global features via cross-attention. Unlike DMAWM, MARIE reconstructs the observation as the input to the policy, forgoing the core benefits of the latent representation, such as history compression and sample efficiency.
> - MABL [B] introduces a bi-level latent-variable world model that factorizes each agent's latent state into a global state and an agent-level state. In contrast, DMAWM directly forces the latent states during imagination to reflect a factorized structure, bypassing the need for the global transition model.
> - CoDreamer [C] extends DreamerV3 to multi-agent settings by adding two levels of GNN-based communication: one within the RSSM world model for state representation, and one within the actor-critic networks for policy coordination. Consequently, CoDreamer relies heavily on inter-agent communication during execution, while DMAWM is designed for entirely communication-free decentralized execution.
>
> > W3, Q2: Missing experimental comparisons with [A], [B], [C]
>
> We completely agree that comparing against these recent and relevant algorithms is crucial for positioning our work. We respectfully clarify that we have already included MARIE [A] as a baseline. Because MARIE is very computationally demanding, we evaluated DMAWM directly on MARIE’s benchmark and followed their experimental protocol to ensure a fair comparison. As reported in Appendix G, DMAWM matches or outperforms MARIE on 10 out of 13 maps.
>
> We address the remaining two of the suggested baselines below:
> - MABL [B]: We encountered execution errors with their provided codebase, so we instead compare against the results reported in their paper. As shown below, DMAWM consistently outperforms MABL across all evaluated SMAC maps:
>
> | Maps | Steps | DMAWM (Ours) | MABL |
> | :--- | :---: | :---: | :---: |
> | 2s vs 1sc | 100k | **95.7(4.3)** | 92.0(7.0) |
> | 3s vs 4z | 100k | **94.8(3.5)** | 83.0(18.0) |
> | 3s vs 5z | 200k | **89.1(5.2)** | 31.0(30.0) |
> | Corridor | 450k | **81.6(14.3)** | 52.0(31.0) |
> | 3s5z vs 3s6z | 450k | **23.7(15.3)** | 19.0(17.0) |
>
> - CoDreamer [C]: As the source code for CoDreamer is not publicly available at this time, we evaluate DMAWM on the Melting Pot environments used in CoDreamer, following their experimental protocol. We normalize returns using the min and max values reported in Table 7 of their paper. Results are shown below:
>
> | Maps | Steps | Return | Normalized return |
> | :--- | :---: | :---: | :---: |
> | Daycare | 500k | 155.6(7.7) | 0.82(0.04) |
> | Cooperative Mining | 500k | 574.7(32.7) | 0.57(0.03) |
> | Collaborative Cooking: Asymmetric | 500k | 2507.6(6.5) | 1.00(0.00) |
> | Collaborative Cooking: Forced | 500k | 748.4(562.9) | 15.27(11.49)$^\dagger$ |
>
> $^\dagger$ Our normalized return exceeds 1 because DMAWM achieves a higher return than the maximum (49.00) reported in their paper.
>
> For the aggregated metric, DMAWM achieves an IQM of 0.91, substantially higher than the 0.48 reported by CoDreamer.

---

> > ### Author Rebuttal · Reviewer_b5QB · 2026-04-02
> >
> > Thank you for the clarification. I apologize for overlooking the discussion of [A], [B], [C] in Section 2 - though I note they were mentioned only briefly, often in single sentences, without explicitly highlighting the similarities and differences with DMAWM. The updated comparisons are satisfactory.
> >
> > MARIE is the closest related baseline, and the comparison results were placed in the Appendix, which reviewers are not obligated to consult. I have now reviewed these. I strongly suggest reporting the MARIE comparison in the main paper.
> >
> > Regarding the relationship with MARIE: the key similarity is the use of a transformer as the centralized multi-agent world model for information fusion. The stated difference is that DMAWM avoids observation reconstruction by operating in latent space. However, I would have found this more compelling if the method avoided trajectory reconstruction entirely (i.e., reconstruction-free objectives), which is known to help discard task-irrelevant details - particularly with high-dimensional observations like images - and avoids expensive high-dimensional decoders. There is a substantial body of literature on this (eg: Bisimulation). As it stands, DMAWM still reconstructs during training. However, I won't press for it, this is just a suggestion, as it can avoid multiple reconstructions you do in Eq 5, which is expensive as the number of agents increase, especially under high-dimensional observations.
> >
> > Turning to the core contribution - the disentangled learning objective - I do find it interesting and elegant. However, this raises a natural question: why is a transformer-based fusion necessary at all? I see that the authors ablate by replacing the transformer-based interaction processor (IP) with an MLP, but I could not find sufficient detail on how this ablation was set up. Could the authors elaborate?
> >
> > More broadly, as stated in the rebuttal: "Our work takes the opposite direction: rather than building more expressive fusion, we show that disentangling the latent states is the critical factor." If disentanglement is the key insight rather than expressive fusion, why can't a simple MLP handle inter-agent interactions - or why not run agent-specific world models independently with local information and make disentangled decisions per agent? The current framing does not convincingly justify the architectural complexity.
> >
> > I remain unconvinced regarding the novelty or impact of the contribution, henceforth.

---

> > > ### Author Response · Authors · 2026-04-04
> > >
> > > We thank the reviewer for the thoughtful reply and suggestions to move the MARIE comparison to the main paper. We will include this in the revised version.
> > >
> > > > Reconstruct-free objective
> > >
> > > We appreciate this point. We agree that reconstruction-free objectives can discard task-irrelevant details by learning compact, value-equivalent representations. Our disentanglement objective serves a different purpose: it constrains the interaction predictor to preserve the decentralized execution constraint during imagination. These two goals are orthogonal and complementary, and a reconstruction-free objective can be used on top of our framework. For example, one could remove the reconstruction loss and leverage gradients back-propagated from the critic to train the world model in DMAWM as in MuZero[1] and TD-MPC[2], retaining our disentanglement constraint while further discarding task-irrelevant information.
> > >
> > > References:
> > > [1] Mastering Atari, Go, Chess and Shogi by Planning with a Learned Model. Nature, 2020.
> > > [2] Temporal Difference Learning for Model Predictive Control. ICML, 2022.
> > >
> > > > Details on the MLP ablation (DMAWM-IP)
> > >
> > > In the DMAWM-IP ablation, we replace the interaction predictor with per-agent MLPs, making imagination fully independent for each agent: $z\_t^i \sim p\_\phi(\cdot\mid h\_t^i)$, $h\_{t+1}^i = f\_\phi(h\_t^i, z\_t^i, a\_t^i)$, $(r\_{t+1}^i, c\_{t+1}^i, m\_{t+1}^i) \sim p\_\phi(\cdot\mid h\_t^i, z\_t^i)$. All other design choices remain identical, isolating the effect of joint transition modeling during imagination.
> > >
> > > As shown in Figure 5(a), DMAWM-IP learns significantly slower than the full DMAWM. Without the interaction predictor, each agent's imaginary trajectory ignores the effects of other agents' actions. Consequently, DMAWM-IP implicitly treats teammates as part of the environment. Because teammates are simultaneously updating their policies, this leads to the well-known non-stationarity problem in independent learning.
> > >
> > > > Why is a transformer-based fusion necessary if disentanglement is the key insight? Why not run agent-specific world models independently with local information and make disentangled decisions per agent?
> > >
> > > We would like to clarify that disentanglement does not mean independence.
> > > Multi-agent dynamics is inherently coupled: agent $i$'s next state could depend on what agent $j$ does.
> > > A multi-agent world model that ignores these dependencies would cause significant inefficiency (shown in the DMAWM-IP ablation).
> > >
> > > In our work, we utilize the interaction predictor to model such inter-agent dependencies $p\_\phi(z\_t^{1:n}\mid h\_t^{1:n})$. However, it can be too expressive for the decentralized constraint during imagination, as spurious correlations could cause an agent's latent state to be affected by an unobservable agent's action (shown in Section 5.4). To this end, we need the disentanglement objective to constrain the interaction predictor to be faithful to the decentralized dynamics.
> > >
> > > > Novelty and impact
> > >
> > > Finally, we respectfully note that our core novelty lies in the disentanglement formulation (the KL alignment objective between the joint prior and factorized posterior), instead of the transformer architecture itself.
> > > This objective enables training a multi-agent world model in the latent space while allowing policies to execute in a fully decentralized manner. The resulting algorithm is both performant and training-efficient, and we believe it will provides a strong baseline for future model-based MARL research.

---

### Official Review · Reviewer_vF8d · 2026-03-15

**Soundness:** 2
**Presentation:** 3
**Significance:** 2
**Originality:** 2
**Overall Recommendation:** 3
**Confidence:** 4

**Summary:**

This paper tries to capture how to learn disentangled and factorized latent representations in a multi-agent framework where the latent space can be formed to capture the underlying world model of the system. It tries to combine existing works on world model learning / learning useful representations in the latent space, to the literature in multi-agent frameworks highlighting that learning world models in MA settings remains difficult due to the intricate interactions between multiple agents, whether in a cooperative or competitive setting.

**Compliance With Llm Reviewing Policy:**

Affirmed.

**Key Questions For Authors:**

As pointed out in the comments : the main question is whether we can demosntrate that the learnt representations that we use from the architecutre, coming from different agents, naturally leads to a compositional and factorized rerpesentation. To claim that this helps in disentanglement of representations, to me is an overclaim - and I would like to see evidence that the hypothesis that the paper provides is indeed true. Can we develop simplistic experimental settings to analyse the learnt representations in the MARL framework, and show that why a baseline would not capture the factorized latents but the proposed approach will do?

**Limitations:**

While the paper tries to tackle an important problem and the experimental results seem to be strong (even if there is no evidence of reproducibility of these results) - I think the core limitation is the overclaim that the architecture proposed and the resulting objectives the paper builds on from dreamer, will naturally lead to learning some factorized or compositional form of representation. Unless clear evidence or proper analysis of this hypothesis is provided, it is unclear to me whether the contribution of the paper is indeed useful. It maybe the experimental results show promise because we are using different encoders that can capture the intricate details different agent observes in the environment - and that may help with latent representations; but claiming that learning the underlying latent states becomes more useful and factorized to me is an overclaim and misleading factor of the paper.

**Strengths And Weaknesses:**

Overall comments :


The paper tries to tackle a key problem of how to learnt world models in a multi-agent setting where different agents are cooperating to solve a given task. It builds on existing works from Dreamer based approaches for world models, extends it to a MARL framework, and proposes a module that can learn the factorized representations from different agents, and unifying the embeddings together to be made useful for control.
Experimental results show that the proposed approach outperforms existing baselines significantly; as demonstrated in some of the standard MARL based evaluation benchmarks.
I like the overall framing of the problem, but I think there are missing pieces to the paper that can be made more clearer to make the contribution more significant. Namely : the paper tries to capture disentangled representations, claiming that in a MARL framework, the latent representations can be convoluted when using different encoders for different agents, making it difficult to fully capture the world model aspects.
It is not clear to me why this is the case though and why we need such an overall architecture? What evidence do we have that the representations are not disentangled as it is? I know in the literature that it is highly likely to be - but the core hypothesis this paper builds on seems to be not well justified?
How can we ensure that even if we learnt representations from different encoders for different agents, the resulting representations will not be convoluted and they are factorized or compositional in some sense? The experimental results seem to demonstrate that this helps, but claiming that the architecture naturally leads to factorized representations seem like an over claim?

---

> ### Author Rebuttal · Authors · 2026-03-31
>
> We thank the reviewer for the thoughtful comments on the justification of our core claims. We address each of these concerns below.
>
> > W1: What evidence do we have that the representations are not disentangled as it is? I know in the literature that it is highly likely to be - but the core hypothesis this paper builds on seems to be not well justified?
>
> We agree that providing concrete evidence of entanglement in existing algorithms is crucial, which is exactly why we dedicated Section 5.4 (Probing latent state disentanglement) to demonstrate this phenomenon.
>
> Specifically, entanglement refers to the unwanted coupling in the latent states of agents during the course of imagination, leading to information leakage. This typically occurs because the multi-agent world model is centralized and the transition is calculated based on the joint states and joint actions of all agents.
> To quantitatively measure this, we designed a probing experiment on the `protoss_5_vs_5` map, where we trained an action predictor to predict teammates' actions solely from its individual latent state transition. High prediction accuracy on unobservable teammates implies entanglement, as the global information is leaked into local latent state via transition modeling.
>
> The results validate our core hypothesis: MAMBA achieves $85.0\\%$ accuracy in predicting the actions of unobservable teammates, demonstrating significant information leakage and entanglement. In contrast, DMAWM achieves $42.3\\%$, performing almost identically to the baseline ($40.0\\%$), whose inputs are raw observations. This directly demonstrates that the representations learnt by MAMBA are entangled, while DMAWM learns more factorized representations.
>
> > W2: How can we ensure that even if we learnt representations from different encoders for different agents, the resulting representations will not be convoluted and they are factorized or compositional in some sense? The experimental results seem to demonstrate that this helps, but claiming that the architecture naturally leads to factorized representations seem like an over claim?
>
> We thank the reviewer for this important point. We acknowledge that claiming the architecture alone leads to factorized representations is imprecise and will revise the manuscript accordingly.
>
> Disentanglement in DMAWM arises from the combination of the architecture and training objective. Architecturally, independent agent modules operate in a strictly decentralized manner based solely on local information (Equation 1), providing the structural foundation for disentanglement. By contrast, methods like MAMBA use communication to update agent latent states, which inherently entangles them. The training objective then mathematically enforces factorization: the KL divergence terms in Equation 4 compel the joint prior $p\_\phi(\hat{z}^{1:n}\_t \mid h^{1:n}\_t)$ to approximate the factorized posterior $\prod\_{i=1}^n q\_\psi(z^i\_t \mid h^i\_t, o^i\_t)$, explicitly aligning the latent states during imagination (joint prior) with latent states during decentralized execution (factorized posterior).
>
> > Q1: Can we develop simplistic experimental settings to analyse the learnt representations in the MARL framework, and show that why a baseline would not capture the factorized latents but the proposed approach will do?
>
> We respectfully point the reviewer to the aforementioned probing experiment in Section 5.4, which is designed to quantitatively measure the extent of disentanglement. By predicting the actions of unobservable teammates, we demonstrate that a baseline (i.e., MAMBA) fails to keep latent states disentangled because its communication channels inherently mix the latent states. Conversely, DMAWM's architecture and its training objective successfully enforce disentanglement to the latent states.
>
> To improve the clarity, we will make sure to mention the results from Section 5.4 earlier in the paper to ensure this core evidence is well informed to the reader.

---

> > ### Author Rebuttal · Reviewer_vF8d · 2026-04-01
> >
> > Thank you to the authors for addressing my comments - especially providing further justifications to section 5.4 to clear out my concerns. I think the related concerns I raised to analyse the learnt representations in the MARL framework are now resolved.
> >
> > However, all the concerns I raised are  partially resolved and I'm unwilling to change the score as of now. This is because the concern I raise about learning factorized or compositional representations probably needs experimental evidence compared to the theoretical justification the authors are giving. I would like to see experimentally if this compositionality holds, which is probably not addressable within the rebuttal timeframe?
> >
> > I am still tending borderline reject for the paper because there are some bold/strong claims which are not demonstrated experimentally and there aren't solid theoretical grounds for them to hold either, other than the theoretical explanations the authors are giving.

---

> > > ### Author Response · Authors · 2026-04-04
> > >
> > > We apologize for missing the experiments on the factorized aspect of the method in our initial response. To address your concern, we designed an intervention experiment that directly evaluate the compositionality of the learned latent states.
> > >
> > > **Experiment design.** Given two episodes $\tau$ and $\tilde{\tau}$, we encode the first $6$ steps as context to obtain two groups of latent states $(h\_0^i, z\_0^i)\_{i=1}^n$ and $(\tilde{h}\_0^i, \tilde{z}\_0^i)\_{i=1}^n$. We then swap one agent's latent state, replacing $(h\_0^j, z\_0^j)$ with $(\tilde{h}\_0^j, \tilde{z}\_0^j)$, and perform imagination for $10$ steps by replaying the recorded actions from the original trajectories, once with the original latent states and once with the swapped latent states. We decode the imaginary latent states into observations and compute the average reconstruction error (MSE) over the $10$ imagination steps for the non-swapped agents. If the latent states are truly factorized, swapping one agent's state would not significantly corrupt the reconstructed observations for other agents.
> > >
> > > **Independent setting.** We first evaluate on a custom task on Multi Particle Environments (MPE) with 3 fully independent agents, which is modified from the `simple_spread` task by disabling collision dynamics and restricting each agent to observe only its own position. Since agents are fully independent, the reconstruction error after swap should be largely unaffected. The results are shown in the following table:
> > >
> > > | Reconstruction error | before swap | after swap | Δ |
> > > |---|---|---|---|
> > > | DMAWM | 0.074 | 0.076 | **0.002** |
> > > | MAMBA | 0.063 | 0.087 | 0.024 |
> > >
> > > DMAWM's little change in the reconstruction error confirms that each agent's latent state is compositional, whereas MAMBA's large difference reveals less compositionality.
> > > We assume this is because MAMBA mixes too much global information into the latent states due to communication, which can be inconsistent with the later swapped latent state when the intervention happens, causing the reconstructed error to significantly increase.
> > >
> > > **Coupled setting.** We further evaluate on the `coop_mining` task, where agents must interact to complete the task. Since this task uses image-based observations, we measure pixel-level MSE. The results are shown in the following table:
> > >
> > > | Reconstruction error | before swap | after swap | Δ |
> > > |---|---|---|---|
> > > | DMAWM | 0.054 | 0.066 | **0.012** |
> > > | MAMBA | 0.062 | 0.096 | 0.034 |
> > >
> > > In this setting, DMAWM's $\Delta$ value is larger than in the independent case. This is expected since the swapped agent could affect the observation of the original agents.
> > > Nonetheless, DMAWM still maintains a relatively small change in the reconstruction errors ($0.012$), while MAMBA exhibits a larger degradation ($0.034$). This demonstrates that even in coupled environments, DMAWM's latent states remain more factorized than those of MAMBA.

---

### Decision · Program_Chairs · 2026-04-30

**Decision:**

Accept (regular)

**Comment:**

This paper proposes DMAWM, a disentangled multi-agent world model that extends the Dreamer framework to decentralized multi-agent control. After the rebuttal and reviewer discussion, the reviewers reached a positive consensus. The initially lowest-scoring reviewer raised their score significantly based on the additional baselines and the 20-agent scalability experiments. The remaining reviewers maintained or reinforced their positive assessments. Concerns about novelty, ELBO tightness, and representation disentanglement were adequately addressed through the discussion.